# Control of seed formation allows two distinct self-sorting patterns of supramolecular nanofibers

Ryou Kubota [1], Kazutoshi Nagao[1], Wataru Tanaka[1], Ryotaro Matsumura[1], Takuma Aoyama[2], Kenji Urayama [2] & Itaru Hamachi [1,3✉]

Self-sorting double network hydrogels comprising orthogonal supramolecular nanofibers have attracted attention as artificially-regulated multi-component systems. Regulation of network patterns of self-sorted nanofibers is considered as a key for potential applications such as optoelectronics, but still challenging owing to a lack of useful methods to prepare and analyze the network patterns. Herein, we describe the selective construction of two distinct self-sorting network patterns, interpenetrated and parallel, by controlling the kinetics of seed formation with dynamic covalent oxime chemistry. Confocal imaging reveals the interpenetrated self-sorting network was formed upon addition of O-benzylhydroxylamine to a benzaldehyde-tethered peptide-type hydrogelator in the presence of lipid-type nanofibers. We also succeed in construction of a parallel self-sorting network through deceleration of seed formation using a slow oxime exchange reaction. Through careful observation, the formation of peptide-type seeds and nanofibers is shown to predominantly occur on the surface of the lipid-type nanofibers via highly dynamic and thermally-fluctuated processes.

[1] Department of Synthetic Chemistry and Biological Chemistry, Graduate School of Engineering, Kyoto University, Katsura, Nishikyo-ku, Kyoto 615-8510, Japan. [2] Department of Macromolecular Science and Engineering, Kyoto Institute of Technology, Matsugasaki, Kyoto 606-8585, Japan. [3] JST-ERATO, Hamachi Innovative Molecular Technology for Neuroscience, Kyoto University, Katsura, Nishikyo-ku, Kyoto 615-8530, Japan. ✉email: ihamachi@sbchem.kyoto-u.ac.jp

In living cells, cytoskeletons, such as actin filaments and microtubules, form orthogonal supramolecular nanofibers through self-sorting phenomena[1,2]. However, recent evidence indicates that the interplay through physical contact among these self-sorted cytoskeletal fibers is also essential for key cellular functions including cell motility and mechano-responsiveness[3,4]. For instance, microtubule polymerization guided by a bundle of actin filaments occurred in migratory cells to regulate the turnover of stress fibers and cell motility[5]. Such precise control of interactions between self-sorted nanofibers (that is, the relative spatial position of these nanofibers) may give us important clues for the design of novel biomimetic soft materials.

Supramolecular hydrogels consist of nanofibrous structures that are formed through self-assembly of low-molecular-weight gelators[6–8]. The elaborate design of hydrogelators allows for construction of stimulus-responsive hydrogels, which are promising scaffolds for drug release matrices and regenerative medicine. Self-sorting double network (SDN) hydrogels composed of orthogonal supramolecular nanofibers have recently attracted considerable attention because they provide rational integration of multiple functions[9–38]. While the control of self-sorting phenomena at the monomer level (self-sorting vs coassembly) was recently achieved[19,24,37], the precise control of network patterns of self-sorted nanofibers remains challenging as it involves a higher hierarchical level of self-sorting events. In addition, evaluation of the relative spatial position of such networks is still extremely difficult. For example, several researchers reported self-sorted donor and acceptor supramolecular nanofibers containing p-n heterojunctions and conducted structural characterization by spectroscopy, TEM and/ or SEM[39–47]. However, the explicit discrimination of the network patterns of the self-sorted nanofibers was impossible by these methods[35,48]. Very recently, Adams and coworkers attempted to control the network structure of self-sorted nanofibers by changing the formation kinetics and investigated the resulting network structure by small-angle neutron scattering and electron paramagnetic resonance spectroscopy[47]. Although valuable, these measurements gave only indirect (ensemble) structural information, which is insufficient to determine the network structure in detail. To date, there are no reliable analytical methods useful to investigate self-sorting network patterns, thus, the factors controlling the network patterns have not yet been clearly addressed.

Herein, we describe the construction of two distinct self-sorting network patterns (interpenetrated and parallel) by controlling the kinetics of seed formation (Fig. 1a). Dynamic covalent oxime chemistry is employed for in situ and kinetically controlled formation of the peptide-type nanofibers by modulating the self-assembled properties of peptide-type hydrogelators. Confocal laser scanning microscopic (CLSM) imaging reveals that the interpenetrated self-sorting network is successfully formed upon addition of an appropriate hydroxylamine compound to a benzaldehyde-tethered peptide-type hydrogelator in the presence of the lipid-type nanofibers. The real-time CLSM imaging allows clear visualization of the self-sorting network formation, revealing there are two different sites for the seed formation, the surface of the lipid-type nanofibers and the interstitial water space, and these are competitive. Inspired by these analyses, we successfully demonstrate that deceleration of the seed formation kinetics using oxime exchange chemistry allows for construction of the parallel self-sorting network through preferential seed formation on the surface of the lipid-type nanofibers. The conversion from interpenetrated to parallel self-sorting network structures is also achieved.

## Results

### Design of a benzaldehyde-tethered peptide-type hydrogelator.

According to pioneering examples where dynamic covalent oxime chemistry was used to form nanofiber and hydrogels[49–64], we designed a new peptide-type hydrogelator, **Ald-F(F)F**, that has a benzaldehyde moiety as a reaction handle at the N-terminus of a self-assembled diphenylalanine sequence (Fig. 1b). Given that the N-terminus is quite sensitive to the self-assembly properties of peptide-type gelators[65], it was expected that the fine tuning of chemical properties of **Ald-F(F)F** through oxime bond formation would allow for in situ nanofiber formation and the resultant hydrogelation. To test our idea, we initially examined the gelation property of **Ald-F(F)F** and the more hydrophobic **BnOx-F(F)F**, which was synthesized by mixing **Ald-F(F)F** and O-benzylhydroxylamine. The slightly opaque hydrogels were formed by heating a suspension of **Ald-F(F)F** or **BnOx-F(F)F** in 100 mM MES (pH 6.0) until they dissolved, followed by cooling to room temperature (Supplementary Figs. 1, 2). The critical gelation concentrations (CGCs) of **Ald-F(F)F** and **BnOx-F(F)F** were determined by the tube inversion method to be 8.6 mM and 1.3 mM, respectively, indicating that **BnOx-F(F)F** is a better hydrogelator than **Ald-F(F)F**. We next investigated in situ oxime formation-induced hydrogelation by addition of O-benzylhydroxylamine to a solution of **Ald-F(F)F** (4.3 mM, below CGC) (Fig. 2a). An opaque hydrogel was formed 1 h after addition of 1 eq of O-benzylhydroxylamine as confirmed by macroscopic sol–gel transition and rheological analysis (Fig. 2b, c, Supplementary Fig. 3). HPLC analysis revealed that 95% of **Ald-F(F)F** was converted to **BnOx-F(F)F** (Fig. 2d). Also, the storage modulus of an **Ald-F(F)F** hydrogel (17.3 mM, above CGC) increased from 1629 to 8554 Pa upon treatment of O-benzylhydroxylamine (Supplementary Fig. 4).

We then analyzed the sol–gel transition by CLSM[66,67]. The fluorescent probe **NP-Alexa647**, which contains a diphenylalanine self-assembled motif similar to **Ald-F(F)F**, was used to stain the peptide-type nanofibers (Fig. 1b)[36]. CLSM imaging of the solution of **Ald-F(F)F** and **NP-Alexa647** demonstrated that the well-entangled nanofibrous structure formed upon addition of O-benzylhydroxylamine, which closely corresponded to the sol–gel transition (Fig. 2e, middle). Time-lapse CLSM imaging demonstrated that the nanofiber formation proceeded stochastically with an induction time of ca. 3 min (Supplementary Fig. 5, Supplementary Movie 1). The results revealed that the nanofibers elongated from small seeds at a velocity of $4 \pm 2\,\mu m/min$ suggesting they followed a nucleation-elongation process (Supplementary Fig. 6)[35,68,69]. As a control, we did not detect any fibrous structure upon addition of buffer solution lacking O-benzylhydroxylamine (Fig. 2e, right).

### Formation of an interpenetrated self-sorting network.

We next attempted in situ fiber formation of **BnOx-F(F)F** in the presence of lipid-type nanofibers (Fig. 3a). As a lipid-type hydrogelator, we used **Phos-MecycC_5**, which shows a good self-sorting ability against a range of peptide-type hydrogelators (Fig. 1b)[37]. The construction of an orthogonal self-sorting network composed of the pair was highly expected because the peptide fiber formation can be temporally controlled when hydroxylamine is added to this system. Indeed, we and others recently revealed that kinetic differentiation of fiber formation between the pair is one of the critical controlling factors for an SDN of supramolecular fibers[18,35]. The formation of peptide-type nanofibers in the presence of the lipid-type nanofibers was observed by CLSM imaging. To selectively stain the peptide- and lipid-type nanofibers, **NP-Alexa647** and **NBD-cycC_6** were used as fluorescent probes, respectively (Fig. 1b) (see Supplementary Fig. 7 for the staining

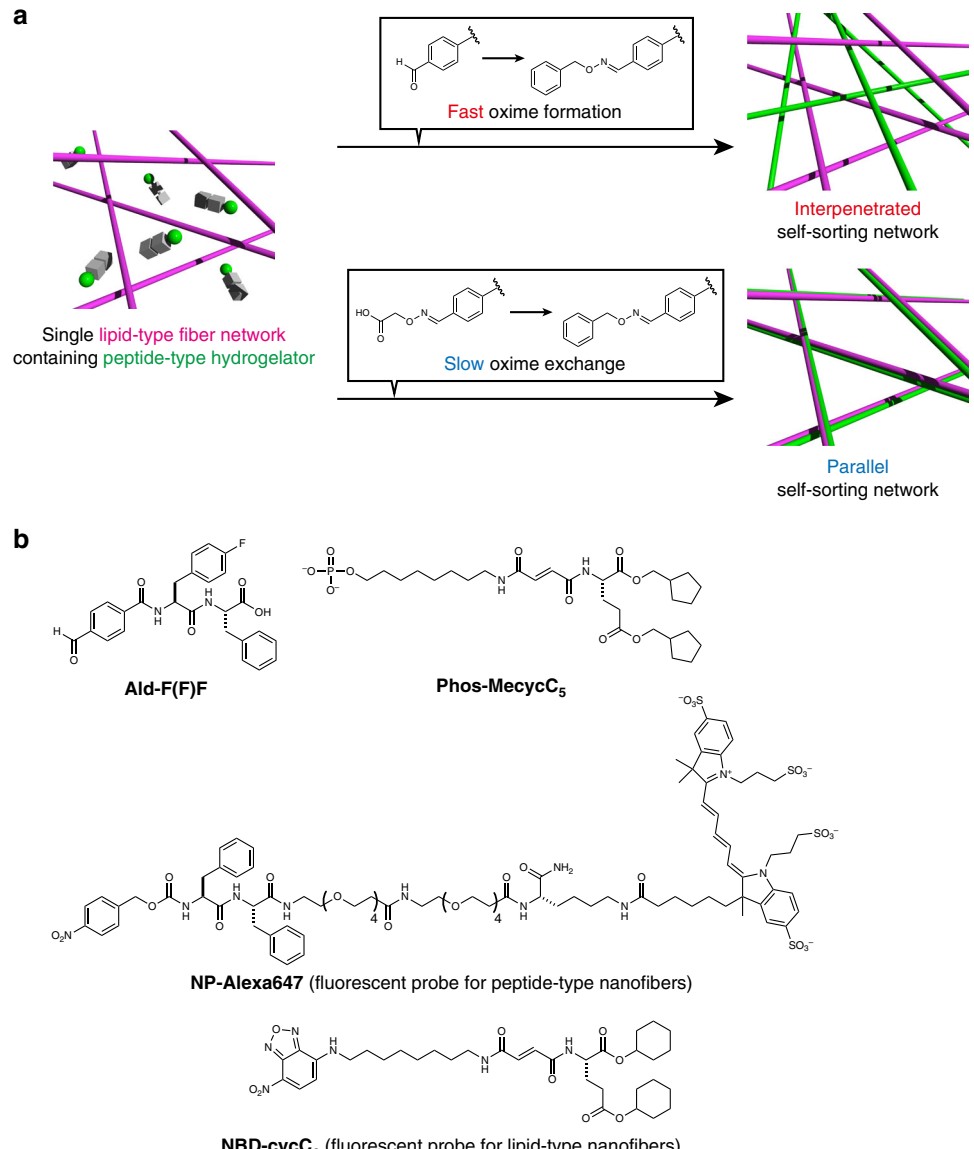

**Fig. 1 Selective formation of interpenetrated and parallel self-sorting networks by dynamic covalent oxime chemistry. a** Schematic illustration of the formation of two distinct network patterns of self-sorted nanofibers by dynamic covalent oxime chemistry. **b** Chemical structures of peptide-, lipid-type hydrogelators (**Ald-F(F)F** and **Phos-MecycC₅**, respectively), and fluorescent probes (**NP-Alexa647** for peptide-type nanofibers and **NBD-cycC₆** for lipid-type nanofibers). See Supplementary Table 1 for abbreviations used in this paper.

selectivity)[36]. We prepared a viscous solution of **Ald-F(F)F** and **Phos-MecycC₅** by heating a suspension of both hydrogelators until they dissolved followed by cooling to room temperature (4.3 and 2.4 mM, respectively; **Phos-MecycC₅** self-assembled into nanofibers under this condition). CLSM imaging of the viscous solution containing **Ald-F(F)F**, **Phos-MecycC₅**, **NP-Alexa647**, and **NBD-cycC₆** allowed visualization of the fibrous structures of **Phos-MecycC₅** stained with **NBD-cycC₆** but not **Ald-F(F)F** (Fig. 3b, top). Upon addition of *O*-benzylhydroxylamine to this solution, the peptide fibers (**BnOx-F(F)F**) stained with **NP-Alexa647** newly appeared and these two fibers formed a self-sorting double network within 1 h (Fig. 3b, bottom). The line plot analysis indicated that the peak tops of the fluorescent intensity of the fibers of **Phos-MecycC₅/NBD-cycC₆** scarcely overlapped with those of **BnOx-F(F)F/NP-Alexa647** fibers, which was also supported by the low Pearson's correlation coefficient (0.14) (Fig. 3c, Supplementary Fig. 9)[70]. 3D stacked imaging showed that the self-sorting double network also formed along the depth direction

(Supplementary Figs. 10, 11, Supplementary Movie 2). HPLC analysis showed that 94% of **Ald-F(F)F** was converted to **BnOx-F(F)F** whereas **Phos-MecycC₅** remained intact, suggesting that **Phos-MecycC₅** showed negligible effects on the oxime formation and the subsequent generation of **BnOx-F(F)F** nanofibers (Supplementary Fig. 12). These results indicated that the interpenetrated SDN was successfully constructed through the oxime formation reaction. It is also confirmed that a range of concentrations of the peptide-, lipid-type hydrogelators, and fluorescent probes scarcely affect formation of the interpenetrated SDN, except for a lower concentration of **Phos-MecycC₅** (Supplementary Figs. 13 and 14).

To support the importance of in situ peptide fiber formation (termed oxime-formation protocol) by dynamic covalent chemistry for the interpenetrated SDN of **BnOx-F(F)F** and **Phos-MecycC₅**, we examined the self-sorting behavior of **BnOx-F(F)F** and **Phos-MecycC₅** by macroscopic observation and rheological analysis. We prepared a mixture of **BnOx-F(F)F** and

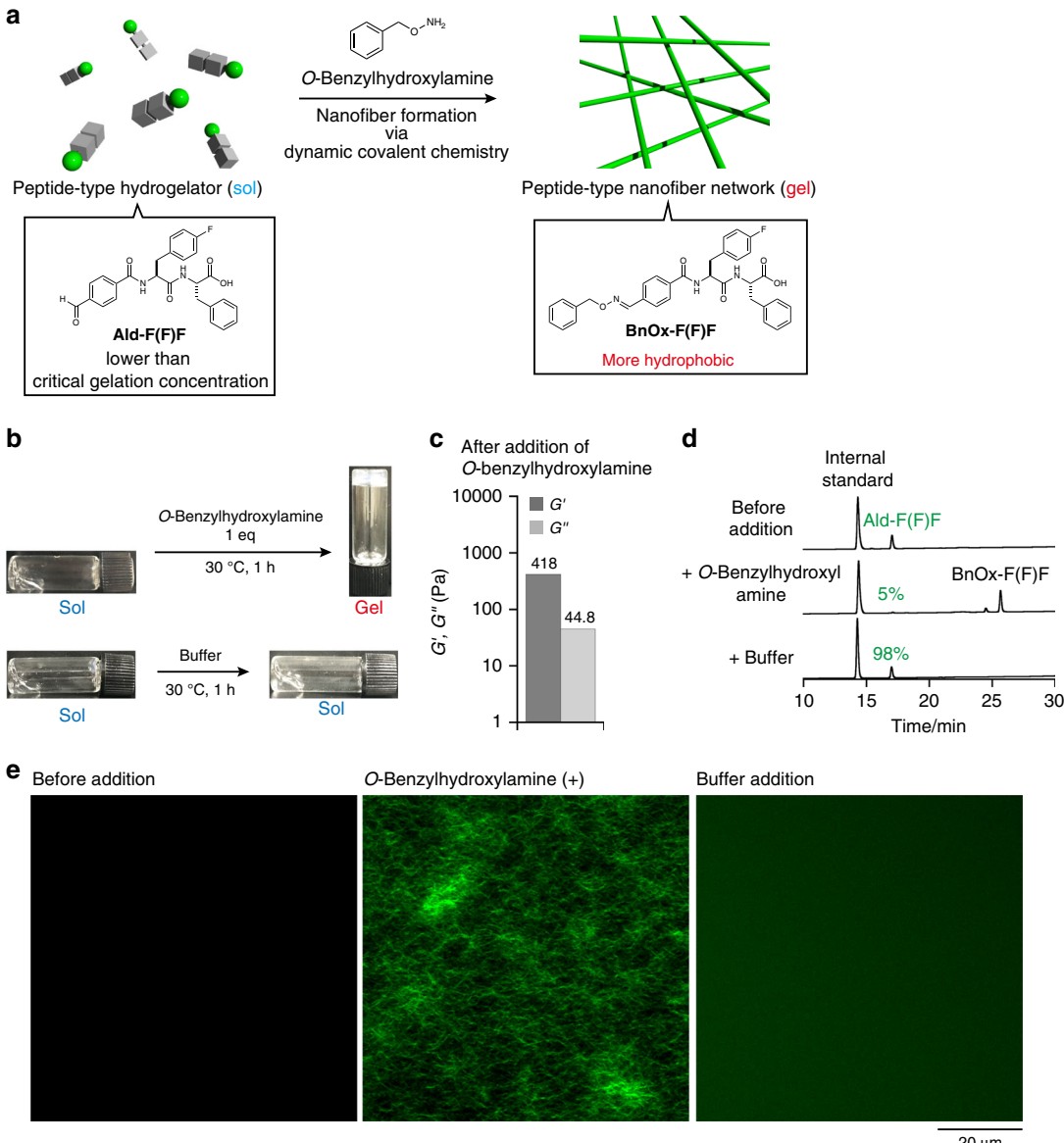

**Fig. 2 Formation of peptide-type nanofibers and hydrogel through an oxime formation reaction. a** Schematic illustration of the nanofiber formation (sol–gel transition) through oxime formation. **b** Macroscopic sol–gel transition of **Ald-F(F)F** upon (top) *O*-benzylhydroxylamine and (bottom) buffer treatment. **c** Rheological properties of the hydrogel. *G'*: storage shear modulus, *G"*: loss shear modulus. Frequency: 10 rad/s, strain amplitude: 1%. **d** HPLC analysis of the formation rate of **BnOx-F(F)F**. Internal standard: fluorescein. **e** High-resolution Airyscan CLSM images (left) before and 1 h after addition of (middle) *O*-benzylhydroxylamine or (right) buffer to **Ald-F(F)F**. Condition: [**Ald-F(F)F**] = 4.3 mM (0.20 wt%), [*O*-benzylhydroxylamine] = 4.3 mM (1.0 eq), [**NP-Alexa647**] = 4.0 μM, 100 mM MES, pH 6.0.

**Phos-MecycC$_5$** by heating until dissolving followed by cooling to room temperature (termed heat-cool protocol). In this case, a suspension containing a white precipitate was formed instead of a hydrogel (Supplementary Fig. 15a), which was in sharp contrast with the in situ oxime-formation protocol that produced a slightly opaque hydrogel by addition of *O*-benzylhydroxylamine to a viscous mixture of **Ald-F(F)F** and **Phos-MecycC$_5$** (Supplementary Fig. 15b). Strain sweep rheological analysis suggested that the hydrogel and the suspension showed distinct rheological properties (Supplementary Figs. 16, 17). The hydrogel had a linear viscoelastic region up to at least 10% strain (Supplementary Fig. 16b), whereas the suspension showed a nonlinear rheological response at more than 0.1% strain (Supplementary Fig. 17a). Notably, the storage modulus of the interpenetrated SDN hydrogel (869 Pa) was higher than the simple sum of the **Phos-MecycC$_5$** solution and **BnOx-F(F)F** gel (32.5 and 418 Pa,

respectively), suggesting that integration of the peptide- and lipid-type networks increased the mechanical property of the hydrogel. Furthermore, CLSM imaging revealed that the suspension consisted of a complex mixture containing spherical aggregates stained with **NBD-cycC$_6$**, peptide-type nanofibers stained with **NP-Alexa647**, and nanofibrous structures stained with both **NP-Alexa647** and **NBD-cycC$_6$** (Fig. 3d). A suspension containing white precipitates was also obtained by mixing hot solutions of **BnOx-F(F)F** and **Phos-MecycC$_5$** that were prepared separately (Supplementary Fig. 18). Such coassembly behavior can be explained by the hydrophobicity of the peptide-type hydrogelator, one of the control factors over self-sorting phenomena we previously found[37]. If the peptide-type hydrogelator is highly hydrophobic, peptide- and lipid-type hydrogelators tend to form coassembled structures, such as spherical aggregates, by the heat-cool protocol. These results clearly

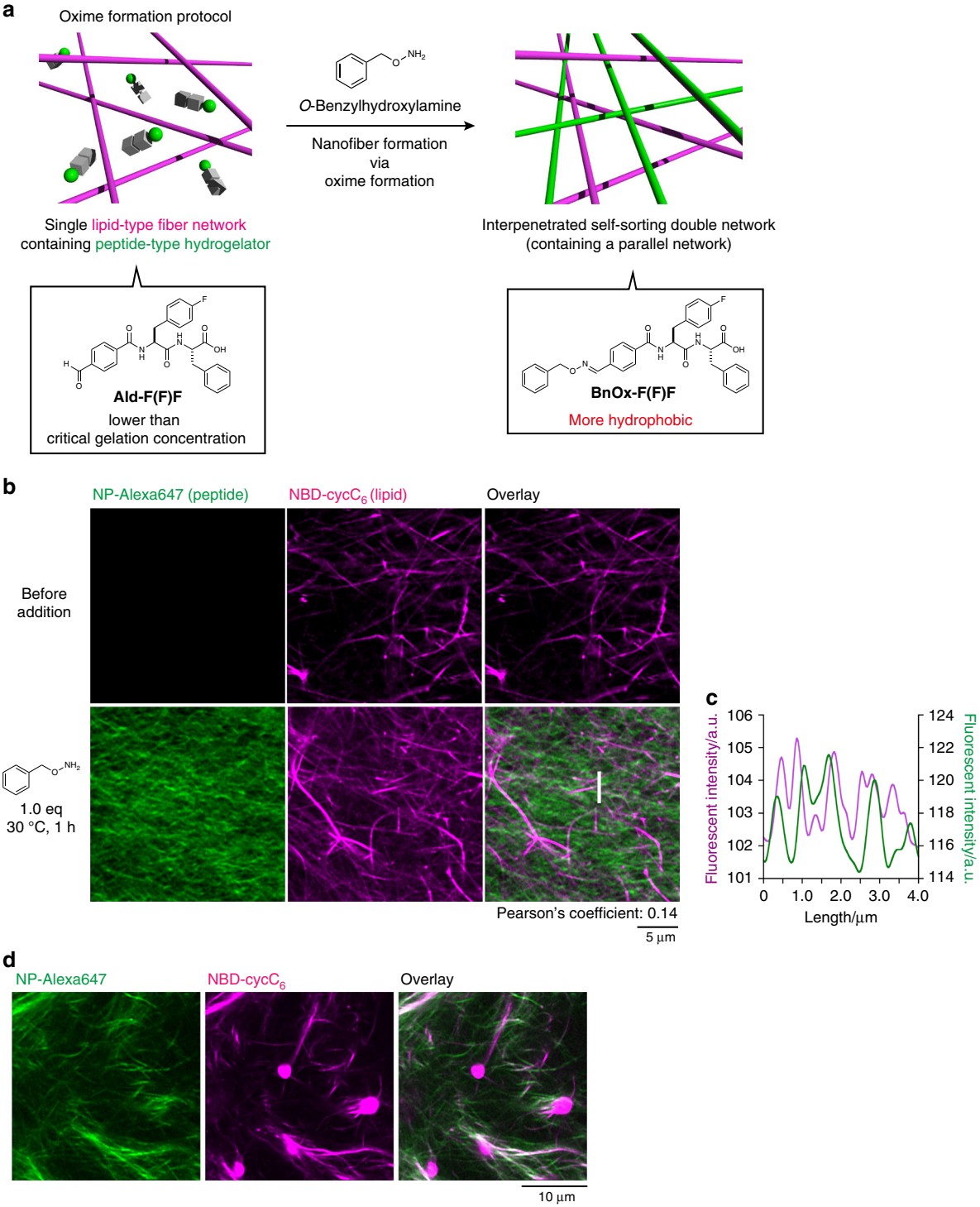

**Fig. 3 Construction of the interpenetrated self-sorting network through the oxime formation protocol. a** Schematic illustration of the construction of an interpenetrated self-sorting double network by the oxime formation protocol. **b** High-resolution Airyscan CLSM imaging (top) before and (bottom) 1 h after addition of *O*-benzylhydroxylamine. The peptide-type nanofibers did not form upon buffer treatment (Supplementary Fig. 8). **c** Line plot analysis along a white line as shown in Fig. 3b. Condition: [**Ald-F(F)F**] = 4.3 mM (0.20 wt%), [**Phos-MecycC₅**] = 2.4 mM (0.15 wt%), [*O*-benzylhydroxylamine] = 4.3 mM (1.0 eq), [**NP-Alexa647**] = 4.0 μM, [**NBD-cycC₆**] = 4.0 μM, 100 mM MES, pH 6.0, 30 °C, 1 h. **d** High-resolution Airyscan CLSM imaging of a suspension of **BnOx-F(F)F**, **Phos-MecycC₅**, **NP-Alexa647**, and **NBD-cycC₆** prepared by a heat-cool protocol. (Left) Alexa647 channel, (middle) NBD channel, (right) overlay image. Green: **NP-Alexa647**, magenta: **NBD-cycC₆**. Condition: [**BnOx-F(F)F**] = 4.3 mM (0.25 wt%), [**Phos-MecycC₅**] = 2.4 mM (0.15 wt%), [**NP-Alexa647**] = 4.0 μM, [**NBD-cycC₆**] = 4.0 μM, 100 mM MES, pH 6.0.

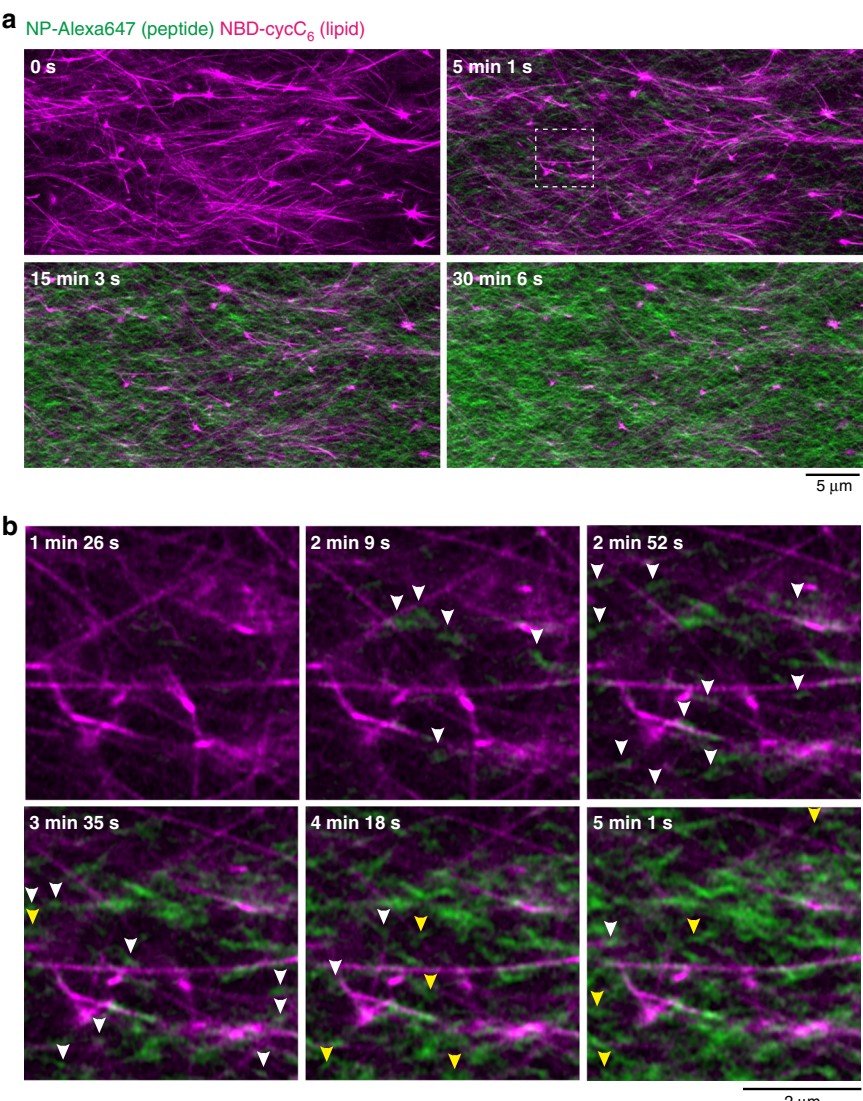

**Fig. 4 Real-time imaging of the formation process of the interpenetrated self-sorting network. a** Time-lapse imaging of the formation of the interpenetrated SDN. **b** Magnified view of the formation of peptide-type nanofibers on the surface of lipid-type nanofibers (highlighted by white arrows) and in the water layer (highlighted by yellow arrows). The white square region in Fig. 4a was magnified. Green: **NP-Alexa647**, magenta: **NBD-cycC₆**. Condition: [**Ald-F(F)F**] = 4.3 mM (0.20 wt%), [**Phos-MecycC₅**] = 2.4 mM (0.15 wt%), [$O$-benzylhydroxylamine] = 4.3 mM (1.0 eq), [**NP-Alexa647**] = 4.0 µM, [**NBD-cycC₆**] = 4.0 µM, 100 mM MES, pH 6.0, 30 °C.

indicated that kinetically controlled formation of the hydrophobic **BnOx-F(F)F** using the in situ oxime formation is crucial for construction of the interpenetrated SDN.

We next attempted to use in situ time-lapse imaging for monitoring the formation process of SDN structure. We added a buffer solution of $O$-benzylhydroxylamine to the solution containing **Ald-F(F)F**, **Phos-MecycC₅**, **NP-Alexa647**, and **NBD-cycC₆** on a glass bottom dish, and then started CLSM imaging. To our surprise, careful real-time observation revealed two distinct processes in the formation of peptide-type nanofibers (Fig. 4a, Supplementary movie 3). Before addition of $O$-benzylhydroxylamine, only long nanofibers of **Phos-MecycC₅** were observed. From 1.5~3.5 min after $O$-benzylhydroxylamine addition, short peptide-type nanofibers formed proximal to the surface of the **Phos-MecycC₅** nanofibers, and elongated beside them (Fig. 4b). After 4 min, many short peptide-type nanofibers stochastically appeared in the interstitial water space surrounded with **Phos-MecycC₅** nanofibers and then elongated randomly to

construct the peptide-type gel network. A larger amount of peptide-type nanofibers were formed in the interstitial water space relative to those proximal to the lipid-type nanofiber surface resulting in the predominant formation of the interpenetrated SDN. These observations implied the nucleation competitively occurred in two distinct locations, that is the surface of the **Phos-MecycC₅** nanofibers and the interstitial water space.

**Parallel SDN formation through a two-step oxime exchange.** According to the in situ CLSM imaging data described above, we envisioned that deceleration of the **BnOx-F(F)F** generation may enable preferential nucleation/elongation proximal to the lipid-type nanofibers leading to construction of a parallel self-sorting network. To suppress the rate of the nucleation process, we sought to use a two-step oxime exchange reaction. As shown in Fig. 5a, we initially added carboxymethoxylamine to an **Ald-F(F)F** hydrogel to produce an oxime peptide, **CaOx-F(F)F**. Since **CaOx-F(F)F** is more hydrophilic at its N-terminus, it should

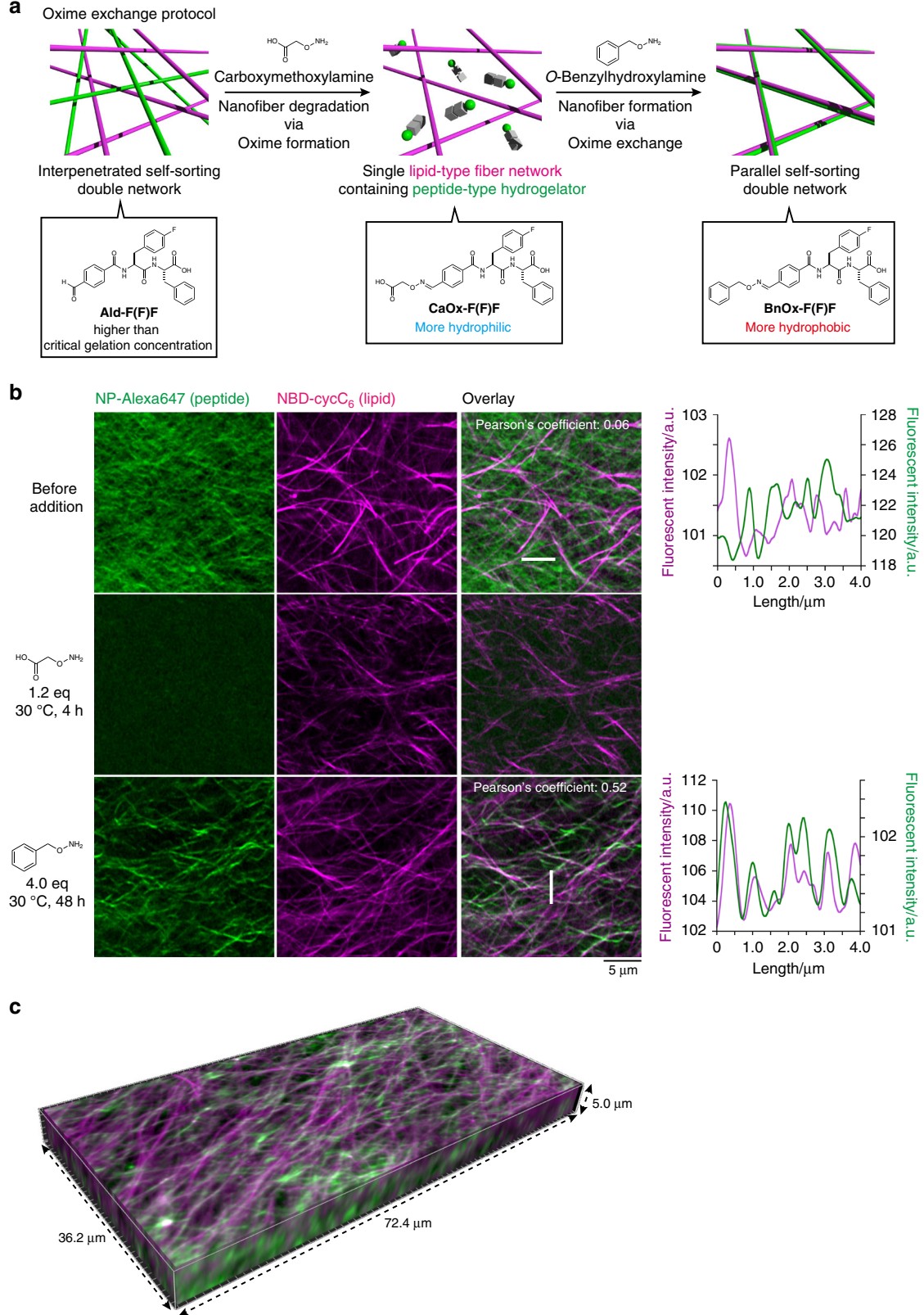

**Fig. 5 Formation of the parallel self-sorting network through the oxime exchange protocol. a** Schematic illustration of the transformation from the interpenetrated SDN to the parallel SDN by the oxime-exchange protocol. **b** High-resolution Airyscan CLSM images of (top) the interpenetrated SDN, (middle) the single lipid network, and (bottom) the parallel SDN. (Left) Alexa647 channel, (second column from the left) NBD channel, (third column from the left) overlay images, (right) line plot analyses along the white lines shown in the overlay images. **c** 3D Airyscan CLSM imaging of the parallel SDN. Green: **NP-Alexa647**, magenta: **NBD-cycC₆**. Condition: [**Ald-F(F)F**] = 17.3 mM (0.80 wt%), [**Phos-MecycC₅**] = 2.4 mM (0.15 wt%), [carboxymethoxylamine] = 20.8 mM (1.2 eq), [O-benzylhydroxylamine] = 69.2 mM (4.0 eq), [**NP-Alexa647**] = 4.0 μM, [**NBD-cycC₆**] = 4.0 μM, 100 mM MES, pH 6.0, 30 °C.

exhibit poor hydrogelation compared with **Ald-F(F)F** according to our previously established design principle[65]. Indeed, the critical gelation concentration of **CaOx-F(F)F** was determined to be 26 mM, which is much higher than that of **Ald-F(F)F** (8.6 mM) (Supplementary Fig. 19, Supplementary Table 2). Subsequently, *O*-benzylhydroxylamine was added to facilitate the formation of peptide-type nanofibers of the superior gelator **BnOx-F(F)F** via an oxime exchange reaction (hereafter we refer to this experiment as the oxime-exchange protocol). We conducted in situ imaging of fiber degradation and formation by the two-step oxime exchange process (see Supplementary Fig. 20 for the staining selectivity). CLSM imaging of the **Ald-F(F)F/Phos-MecycC$_5$** hydrogel stained with **NP-Alexa647** and **NBD-cycC$_6$** revealed that **Ald-F(F)F** and **Phos-MecycC$_5$** nanofibers formed the interpenetrated SDN structure (Fig. 5b, top). The line plot analysis showed that the peak tops of peptide- and lipid-type nanofibers were not overlapped but alternated with each other, and the colocalization analysis indicated no correlation between the two nanofibers (Pearson's correlation coefficient: 0.06) (Supplementary Fig. 21). The 3D image demonstrated that the interpenetrated SDN structure was also formed along the depth direction (Supplementary Figs. 22, 23, Supplementary Movie 4). These results indicated that **Ald-F(F)F** and **Phos-MecycC$_5$** are a good self-sorting pair. Upon addition of carboxymethoxylamine (1.2 eq) to this self-sorting network, the peptide-type nanofibers completely disappeared in 4 h, whereas **Phos-MecycC$_5$** nanofibers were almost unchanged (Fig. 5b, middle). Notably, the peptide-type nanofibers formed again 48 h after addition of *O*-benzylhydroxylamine (4.0 eq) and their localization appeared to overlap with the **Phos-MecycC$_5$** nanofibers (Fig. 5b, bottom). The Pearson's correlation coefficient was determined to be 0.52, which was greater than the original interpenetrated SDN, implying the peptide- and lipid-type nanofibers were moderately correlated in their spatial localization. The line plot analysis demonstrated that the peak tops of peptide- and lipid-type nanofibers were not completely overlapped but slightly out of alignment (98 ± 80 nm), while the overall peak patterns were quite similar to each other (Fig. 5b, bottom right, Supplementary Fig. 24). The average peak-top distance of the parallel SDN was statistically smaller than those of the interpenetrated SDNs of **BnOx-F(F)F/Phos-MecycC$_5$** and of **Ald-F(F)F/Phos-MecycC$_5$** (see Supplementary Fig. 25 for statistical analysis). The 3D imaging revealed that a similar network structure also formed along the depth direction (Fig. 5c, Supplementary Fig. 26, Supplementary Movie 5). These observations demonstrated that peptide- and lipid-type hydrogelators formed the parallel SDN rather than the coassembled structure.

We further analyzed the oxime-exchange process by HPLC analysis and the macroscopic phase transition. HPLC analysis showed that the first oxime formation (by carboxymethoxylamine) and the second oxime exchange (with *O*-benzylhydroxylamine) proceeded in 98 and 30% yield, respectively, whereas **Phos-MecycC$_5$** remained unchanged (Supplementary Fig. 27). The conversion ratio of **Ald-F(F)F** to **BnOx-F(F)F** in the second step (4.9 mM) was higher than the CGC of **BnOx-F(F)F** (1.3 mM), and the resultant concentration of **BnOx-F(F)F** was comparable with that in the oxime-formation protocol (4.3 mM). As controls, buffer treatment at each step induced neither collapse nor formation of the peptide-type nanofibers (Supplementary Fig. 28, 29). We also confirmed that addition of carboxymethoxylamine followed by *O*-benzylhydroxylamine induced the macroscopic gel–sol–gel transition, which was consistent with rheological analysis (Supplementary Figs. 30, 31).

To examine requirements for constructing the parallel SDN, we tested different concentrations and preparation protocols by CLSM imaging. The parallel SDN structure was successfully formed under different concentrations of **Ald-F(F)F**, **Phos-**

**MecycC$_5$**, and *O*-benzylhydroxylamine (Supplementary Fig. 32). The initial in situ formation of **CaOx-F(F)F** was essential for formation of the parallel SDN; a suspension with white precipitate was formed by the direct mixing of **CaOx-F(F)F** and **Phos-MecycC$_5$** with the heat-cool protocol (Supplementary Fig. 33). To decelerate the kinetics of seed formation in the oxime-formation protocol, we attempted stepwise addition of *O*-benzylhydroxylamine to a viscous solution of **Ald-F(F)F** and **Phos-MecycC$_5$**. In this case, however, the interpenetrated SDN was mainly formed (Supplementary Fig. 34). Also, CLSM imaging of the sample prepared (a heterogeneous precipitate thus obtained) by slowly cooling down the hot solution of **BnOx-F(F)F** and **Phos-MecycC$_5$** showed the mixed network structure of interpenetrated and parallel SDNs (Supplementary Fig. 35). Moreover, the direct treatment of the interpenetrated SDN of **Ald-F(F)F** and **Phos-MecycC$_5$** with *O*-benzylhydroxylamine never induced transformation to the parallel SDN (Supplementary Figs. 36, 37). It was also confirmed that the further conversion from **BnOx-F(F)F** to **CaOx-F(F)F** did not proceed upon addition of an excess amount of carboxymethoxylamine, implying that the self-assembled structure stabilizes **BnOx-F(F)F** probably due to its tight packing and/or the hydrophobic microenvironment (Supplementary Fig. 38). These experiments indicated that the two-step oxime exchange was essential for construction of the parallel SDN.

**Real-time CLSM imaging of the parallel SDN formation.** The time-lapse CLSM imaging of the formation process revealed that the generation of short peptide-type nanofibers was exclusively initiated proximal to the **Phos-MecycC$_5$** fibers 1 h 40 min after *O*-benzylhydroxylamine addition, and the fibers elongated slowly along the **Phos-MecycC$_5$** nanofibers (Fig. 6a, b, Supplementary Movie 6). In total, 98.6% of peptide-type seeds (277 out of 281) were formed on the surface of the lipid-type nanofibers and elongated along the lipid-type nanofibers. Importantly, the elongating fibers never completely merged with the **Phos-MecycC$_5$** fibers during these processes. Moreover, we identified a variety of unique nanofiber elongation/collapse events from the real-time CLSM movie. As shown in Fig. 7a, the velocity of nanofiber elongation was stochastically changed. A nanofiber grew 1.6 μm from the seeds during the initial 20 min (average velocity: 0.08 μm/min), whereas the average velocity decreased to 0.05 μm/min during the next 20 min. After the elongation stopped for 1 h 40 min, this nanofiber again elongated by 0.02 μm/min along the direction of a **Phos-MecycC$_5$** fiber. A connection between two fragments of peptide-type nanofibers was observed as shown in Fig. 7b. Two seeds formed proximal to the same **Phos-MecycC$_5$** nanofiber surface at 1 h 40 min, and these peptide-type nanofibers grew independently from the seeds for 20 min. Elongation of both nanofibers slowed almost to termination over the next 40 min followed by connection of the two nanofibers. Figure 7c shows formation of a branched peptide-type nanofiber. During elongation, the peptide-type nanofiber appeared to interact with the terminus of another nanofiber resulting in branching point formation. It was also found that a peptide-type nanofiber transferred from a parallel **Phos-MecycC$_5$** nanofiber to another during elongation as shown in Fig. 7d. Surprisingly, a few peptide-type nanofibers gradually collapsed (depolymerization) during the time-lapse imaging (Fig. 7e). The imaging data provided direct evidence that the formation of the artificial SDN structure is very dynamic and subject to thermal fluctuations[71].

On the basis of the real-time CLSM observations, it was clear that there were critical differences in the kinetics and spatial locations of the peptide-type nanofiber formation between the

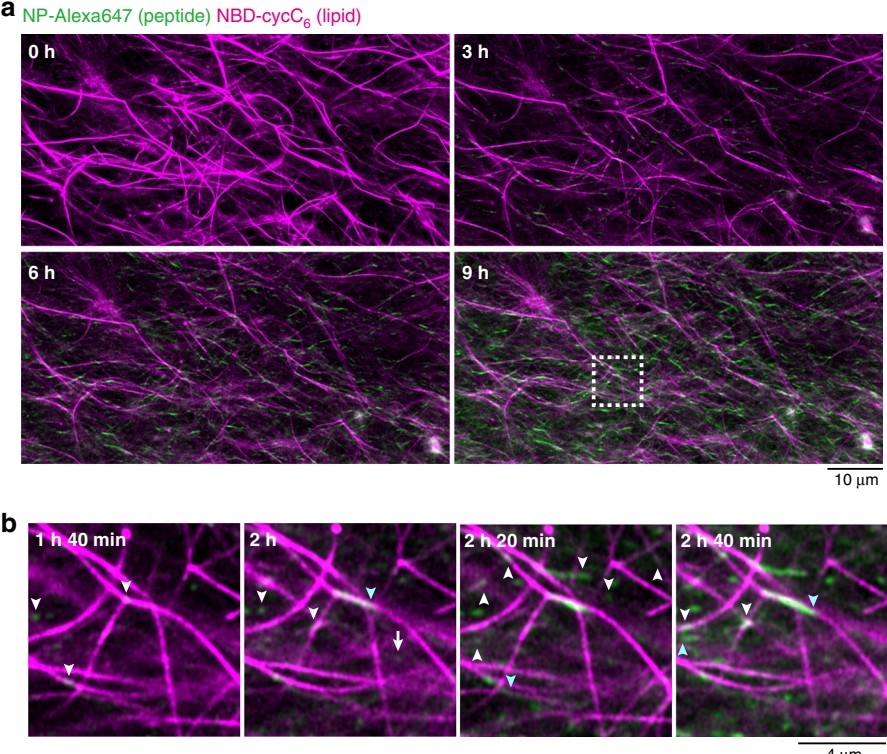

**Fig. 6 Real-time imaging of formation process of the parallel self-sorting network. a** Time-lapse imaging of the formation process of the parallel SDN.
**b** Magnified view of seed formation on the surface of lipid-type nanofibers (a white square in Fig. 6a). White and sky blue arrowheads represent the sites of
seed formation and elongation on the surface of lipid nanofibers, respectively. Green: **NP-Alexa647**, magenta: **NBD-cycC_6**. Condition: [**Ald-F(F)F**] = 17.3 mM
(0.80 wt%), [**Phos-MecycC_5**] = 2.4 mM (0.15 wt%), [carboxymethoxylamine] = 20.8 mM (1.2 eq), [O-benzylhydroxylamine] = 69.2 mM (4.0 eq),
[**NP-Alexa647**] = 4.0 µM, [**NBD-cycC_6**] = 4.0 µM, 100 mM MES, pH 6.0, 30 °C.

oxime-formation and oxime-exchange protocols. In the oxime-formation protocol, the short peptide-type nanofibers appeared 1.5 min after addition of O-benzylhydroxylamine, whereas it took at least 1 h 40 min in the oxime-exchange protocol (Figs. 4b, 6b, respectively). Moreover, the average elongation rates determined by CLSM imaging were 3 ± 3 µm/min and 0.06 ± 0.03 µm/min in the oxime-formation and oxime-exchange protocols, respectively (Supplementary Fig. 40). These results clearly demonstrated that the oxime exchange process substantially slowed down both the nucleation and elongation of the peptide-type nanofibers. Moreover, our CLSM observation clarified the formation site of the peptide-type seeds/nanofibers. In the oxime-exchange protocol, the peptide nanofibers predominantly formed proximal to the surface of the **Phos-MecycC_5** nanofibers, and the formation from the interstitial water space was only minimally observed. This is in sharp contrast with the oxime-formation protocol where the peptide-type nanofibers formed at both sites. Our results suggested that the nucleation barrier on the surface of the **Phos-MecycC_5** nanofibers may be slightly smaller than that in the water space although these sites were competitive. In the oxime-formation protocol, the reaction system should reach a super-saturated state in a short time because of the fast generation kinetics of **BnOx-F(F)F**, therefore the nucleation proceeded in both sites. In addition, the slower kinetics of **BnOx-F(F)F** generation by the oxime exchange allowed for preferential nucleation on the energetically-favorable surface of the **Phos-MecycC_5** nanofibers. These behaviors seem to be consistent with the physics of nucleation barriers as recently demonstrated by liquid droplet formation in live cells with optogenetics[72,73]. The present example demonstrates that controlling the nucleation

process would be a promising strategy to construct an artificially regulated network pattern of supramolecular hydrogels.

We monitored the stability of the interpenetrated and parallel SDNs by their time-dependent changes of CLSM images. The interpenetrated SDN of **BnOx-F(F)F/Phos-MecycC_5** (prepared by the oxime-formation protocol) and **Ald-F(F)F/Phos-MecycC_5** did not show any significant changes at least for 3 days (Supplementary Fig. 41a, b). In contrast, the parallel SDN gradually collapsed to form coassembled spherical aggregates after 3 days (Supplementary Fig. 41c). Interestingly, we found that the higher concentration of the peptide- or lipid-type hydrogelators suppressed collapse of the parallel SDN (Supplementary Fig. 41d, e). These data suggested that the parallel SDN is the kinetically-trapped state and that the stability of the peptide- and lipid-type nanofibers is critical for the kinetic trap of the parallel SDN.

## Discussion
Our in situ imaging-based approach provides the basis for the development of a promising method to control self-sorting supramolecular network patterns. That is, the regulation of seed formation kinetics and sites allowed us to selectively prepare interpenetrated and parallel self-sorting network structures. We achieved visualization of parallel SDN and conversion from interpenetrated to parallel SDN structures. Our strategy to regulate the nucleation processes is one promising method for controlling a hydrogel network at a higher hierarchical level. We believe that our findings will facilitate the rational design of multicomponent supramolecular materials with

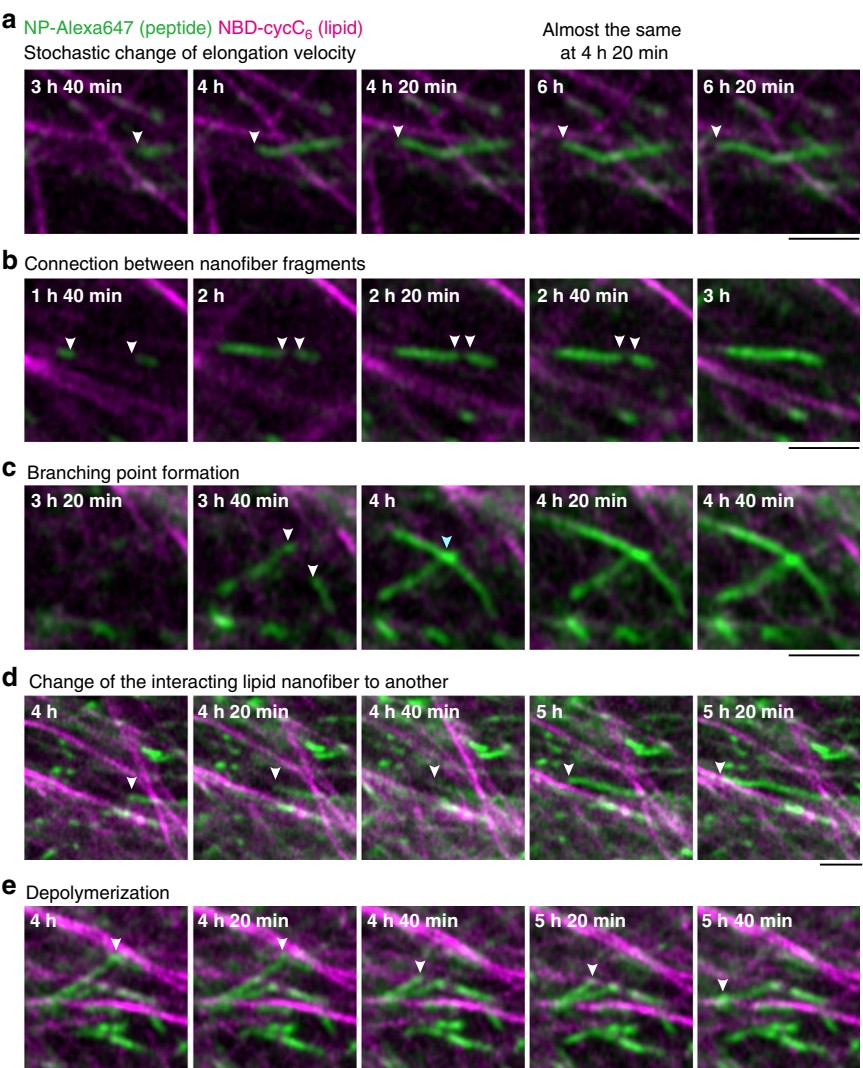

**Fig. 7 Unique elongation/collapse behaviors of the peptide-type nanofibers during the oxime-exchange reaction. a** Stochastic change of elongation velocity, (**b**) connection between nanofiber fragments, (**c**) formation of a branching nanofiber, (**d**) change of interacting lipid-type nanofiber to another, and (**e**) depolymerization. White and sky blue arrows show the tips of peptide-type nanofibers and a crosslinking point, respectively. The detailed location of each magnified image is shown in Supplementary Fig. 39. Scale bar: 2 μm. Green: **NP-Alexa647**, magenta: **NBD-cycC₆**. Condition: [**Ald-F(F)F**] = 17.3 mM (0.80 wt%), [**Phos-MecycC₅**] = 2.4 mM (0.15 wt%), [carboxymethoxylamine] = 20.8 mM (1.2 eq), [O-benzylhydroxylamine] = 69.2 mM (4.0 eq), [**NP-Alexa647**] = 4.0 μM, [**NBD-cycC₆**] = 4.0 μM, 100 mM MES, pH 6.0, 30 °C.

hierarchical-organized structures useful for next-generation, adaptive soft materials.

## Methods

**Preparation of Ald-F(F)F and BnOx-F(F)F hydrogels.** A suspension of an **Ald-F (F)F** or **BnOx-F(F)F** powder in 100 mM MES, pH 6.0 was heated by a heating gun (PJ-206A1, Ishizaki) until dissolving. The resultant hot solution was cooled to room temperature (rt) and incubated for 24 h. The state (gel or sol) of the sample was judged by the tube inversion test. The assay conditions were referred in the figure captions.

**Hydrogelation through in situ formation of BnOx-F(F)F.** A suspension of an **Ald-F(F)F** powder (4.3 mM) in 100 mM MES, pH 6.0 was heated by a heating gun until dissolving. The resultant hot solution was cooled to rt and incubated for 1 h. To this resultant solution, a solution of O-benzylhydroxylamine (43 mM in 100 mM MES, pH 6.0) was added and incubated for 1 h. The state (gel or sol) of the sample was judged by the tube inversion test. The assay conditions were referred in the figure captions.

**In situ formation of BnOx-F(F)F from Ald-F(F)F gel.** A suspension of an **Ald-F (F)F** powder (17.3 mM) in 100 mM MES, pH 6.0 was heated by a heating gun until dissolving. The resultant hot solution was cooled to rt and incubated for 1 h. To this resultant hydrogel, a 10× solution of O-benzylhydroxylamine (173 mM in 100 mM MES, pH 6.0) was added and incubated for 1 h. The hydrogel was analyzed by rheological and HPLC analyses.

**CLSM imaging in the oxime-formation protocol.** The suspension of **Ald-F(F)F** (4.3 mM) and **NP-Alexa647** (4.0 μM) with/without **Phos-MecycC₅** (2.4 mM) and **NBD-cycC₆** (4.0 μM) in 100 mM MES, pH 6.0 was heated by a heating gun until dissolving. After cooling to rt, the resultant mixture (18 μL) was transferred to a glass bottom dish (Matsunami) and incubate at rt for 1 h in the presence of water to avoid dryness. To the resultant solution, a solution of O-benzylhydroxylamine (43.2 mM, 1 μL in 100 mM MES, pH 6.0) or buffer was added. After incubation for 1 h, CLSM imaging was conducted.

**Preparation of the suspension of BnOx-F(F)F and Phos-MecycC₅.** The suspension of **BnOx-F(F)F** (4.3 mM) and **Phos-MecycC₅** (2.4 mM) with/without **NP-Alexa647** (4.0 μM) and **NBD-cycC₆** (4.0 μM) in 100 mM MES, pH 6.0 was heated by a heating gun until dissolving. The resultant mixture was cooled to rt and incubated at rt for 24 h. The state (gel or sol) of the sample was judged by the tube

inversion test. The obtained suspension (containing fluorescent probes) was moved to a glass bottom dish and observed by CLSM imaging.

**Mixing hot solutions of BnOx-F(F)F and Phos-MecycC$_5$.** The suspensions of **BnOx-F(F)F/NP-Alexa647** (8.6 mM and 8.0 μM) and **Phos-MecycC$_5$/NBD-cycC$_6$** (4.8 mM and 8.0 μM) were separately heated by a heating gun until dissolving. The equal volume of the resultant hot solutions were immediately mixed and incubated at rt for 1 h. The state (gel or sol) of the sample was judged by the tube inversion test. The obtained suspension (containing fluorescent probes) was moved to a glass bottom dish and observed by CLSM imaging.

**CLSM imaging in the oxime-exchange protocol.** The suspension of **Ald-F(F)F** (17.3 mM) and **NP-Alexa647** (4.0 μM) with **Phos-MecycC$_5$** (2.4 mM) and **NBD-cycC$_6$** (4.0 μM) in 100 mM MES, pH 6.0 was heated by a heating gun until dissolving. After cooling to rt, the resultant mixture (10 μL) was transferred to a glass bottom dish (Matsunami) before gelation. After incubation at rt for 1 h in the presence of water to avoid dryness, CLSM imaging was conducted. To the resultant hydrogel, a solution of carboxymethoxylamine (208 mM, 1 μL in 100 mM MES, pH 6.0) or buffer was added. After incubation for 4 h, CLSM imaging was conducted. Subsequently, O-benzylhydroxylamine (300 mM, 2.3 μL in 100 mM MES, pH 6.0) or buffer was added to the resulting solution. After incubation at rt for 48 h, CLSM imaging was conducted.

**Suspension of CaOx-F(F)F and Phos-MecycC$_5$.** The suspension of **CaOx-F(F)F** (17.3 mM) and **Phos-MecycC$_5$** (2.4 mM) with/without **NP-Alexa647** (4.0 μM) and **NBD-cycC$_6$** (4.0 μM) in 100 mM MES, pH 6.0 was heated by a heating gun until dissolving. The resultant mixture was cooled to rt and incubated at rt for 1 h. The state (gel or sol) of the sample was judged by the tube inversion test. The obtained suspension (containing fluorescent probes) was moved to a glass bottom dish and observed by CLSM imaging.

**Trial for conversion from BnOx-F(F)F to CaOx-F(F)F.** To a solution of **Ald-F(F)F** (4.3 mM) with/without **NP-Alexa647** (4.0 μM) were added a 10× solution of O-benzylhydroxylamine (43 mM). The resultant mixture was incubated at 30 °C for 1 h. A 10× solution of carboxymethoxylamine (430 mM) was added to the hydrogel and incubated at 30 °C for 24 h. The resulting samples were analyzed by HPLC analysis and CLSM imaging. The state (gel or sol) of the sample was judged by the tube inversion test.

**Slow cooling down of BnOx-F(F)F and Phos-MecycC$_5$ mixture.** The suspension of **BnOx-F(F)F** (4.3 mM) and **Phos-MecycC$_5$** (2.4 mM) with/without **NP-Alexa647** (4.0 μM) and **NBD-cycC$_6$** (4.0 μM) were heated by a heating gun until dissolving. The resultant hot solutions were incubated on a heating plate at 100 °C for 5 min, then the setting temperature was changed from 100 °C to 30 °C, and the sample was incubated for 30 min on the heating plate. The state (gel or sol) of the sample was judged by the tube inversion test. The sample was analyzed by CLSM imaging.

**HPLC analysis of the hydrogels.** The preparation of the hydrogels and solutions was described above. The samples (133 μL) were dissolved by cold DMF (800 μL) and a DMSO solution of fluorescein (internal standard, 3 mM, 90 μL). The resultant mixture was filtered with membrane filter (diameter: 0.45 μm), and then analyzed by RP-HPLC (column: YMC-Triart C18, A:B = 30:70 to 90:10 for 30 min, A: CH$_3$CN containing 0.1% TFA, B: H$_2$O containing 0.1% TFA).

**Rheological analysis.** The preparation of the hydrogel was the same as described above. The resultant disk-shaped hydrogels (ca. 10 mm) were carefully taken out from the PDMS mold and put onto the stage of a rheometer (MCR-502, Anton Paar) with a parallel plate geometry. Strain sweep data were obtained using shear mode at a frequency of 10 rad/s, and linear dynamic viscoelasticity were measured in shear mode at 0.3 or 1.0% strain amplitude for frequency sweep.

**Determination of nanofiber elongation velocity.** The elongation distance of the peptide-type nanofiber was estimated by comparing the successive two images of the time-lapse imaging with Fiji. The velocity was calculated by dividing the elongation distance by interval of time-lapse imaging. In total, 50 and 120 elongated nanofibers were randomly selected (Supplementary Figs. 6 and 40, respectively). The histograms were depicted by using Kaleidagraph 4.5 (Synergy Software).

## Data availability
The authors declare that the data supporting the findings of this study are available with the paper and its Supplementary information files. The data that support the findings of this study are available from the corresponding author upon reasonable request.

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

## Acknowledgements

This work was supported by a Grant-in-Aid for Scientific Research on Innovative Areas "Chemistry for Multimolecular Crowding Biosystems" (JSPS KAKENHI Grant JP17H06348), JST ERATO Grant Number JPMJER1802 to I.H., by a Grant-in-Aid for Young Scientists (JSPS KAKENHI Grant JP18K14333, JP20K15400) to R.K., and by a Research Fellowship from the Japan Society for the Promotion of Science (JSPS) for Young Scientists to W.T. (JSPS KAKENHI Grant JP19J14474). We thank Renee Mosi, PhD, from Edanz Group (http://www.edanzediting.com/ac) for editing a draft of this manuscript.

## Author contributions

I.H. and R.K. conceived the project. R.M. identified that the benzaldehyde hydrogelators showed the macroscopic sol–gel transition upon addition of the hydrophobic hydro-xylamine. K.N. conducted all of the experiments and analyzed the data with I.H., R.K., and W.T. Rheological measurements were conducted by K.N., T.A., and K.U. I.H. and R.K. wrote the manuscript and edited it with assistance from all authors.

## Competing interests

The authors declare no competing interests.
