## [Peer Review File · Nature Communications]

REVIEWER COMMENTS

Reviewer #1 (Remarks to the Author):

REVIEW REPORT FOR:

Control of seed formation allows two distinct self-sorting patterns of supramolecular nanofibers

In this manuscript, Ryou Kubota et al., describe a self-sorting double network hydrogels formation comprising orthogonal nanofibers peptide-type hydrogelator and lipid-type nanofibers. The authors describe that they can obtain two kind of self-sorting network, an interpenetrated network upon the addition of O-benzylhydroxylamine to a benzaldehyde-tethered peptide in the presence of the lipid-type nanofibers; and a parallel network upon a slow oxime exchange reaction from an oxime non-hydrogelator peptide-type to an oxime peptide-type hydrogelator in presence of the lipid-type hydrogelator. In the second case, the lipid-type hydrogelator play the role of seeding to obtain a parallel network formation.

The work claim novelty upon the first direct visualization of self-sorted network structures by using CLSM and the first control of pattern in situ of the network by slowing down the seeding kinetic through the use of oxime exchange chemistry to go from interpenetrated network to a parallel nanofibers network but it is rather preliminary for publication and the authors are hand-wavy the data assessment regarding the parallel network formation.

Conceptually, it may be valuable for Nat. Commun., however, I suggest to address the following comments:

Detailed comments on the text:

Question 1: Page 3, line 14: "drug release matrices, drug delivery".

The two of them mean exactly the same thing.

Question 2: The Ald-F(F)F form a gel on its own and the BnOx-F(F)F too, do the formation in situ of the BnOx-F(F)F change anything in the mechanical properties of the hydrogel? Also, do the BnOx-F(F)F hydrogel have similar hardness compare to the Ald-F(F)F hydrogel? How then the addition of a second self-sorted fibrous network changes the mechanical properties? Does the hydrogel soften or get harder?

Question 3: Can you also control the pattern in the other way around, when the network of fibers is parallel can you transform it back to an interpenetrated one? Does it influence the hydrogel properties (mechanical or others)?

Question 4: Page 8, line 4: "In this case, a suspension containing a white precipitate was formed instead of a hydrogel (supplementary Fig. 11a), which was in sharp contrast with the in situ oxime-formation protocol that produced a slightly opaque hydrogel by addition of BHA to a viscous mixture of Ald-F(F)F and Phos-MecycC5 (supplementary Fig. 11b)."

What happens if you prepare the hydrogel by heating the two hydrogelators separately and then mix them together as in page 5 line 15 you seem to succeed to obtain a hydrogel for Ald-F(F)F and BnOx-F(F)F by heating the solution and let it cool down. Can you explain why it works when you prepare an interpenetrated hydrogel from Ald-F(F)F and Phos-MecycC5 (like for the oxime exchange protocol with the conditions [Ald-F(F)F] = 17.3 mM (0.80 wt%), [Phos-MecycC5] = 9 2.4 mM (0.15 wt%)) and not when you mix BnOx-F(F)F and Phos-MecycC5 ?

Question 5: Page 8, line 18: “dynamic covalent oxime chemistry”.

What do you mean by dynamic? As your reaction is not auto-reversible how can you explain it? If it's for the CLSM pictures showing the growth of fibers (Fig. 6, 7 and S5) can it be only dependent of the slow oxime exchange and slow growth of the fiber network, which may need several hours to reach it's steady-state as the two chemical processes are kinetically different?

Question 6: Page 12, line 14: “Surprisingly, a few peptide-type nanofibers gradually collapsed (depolymerization) during the time-lapse imaging (Fig. 7e). The imaging data provided direct evidence that the formation of the artificial SDN structure is very dynamic and subject to thermal fluctuations”.

Does that mean that after some time the whole fiber network collapses? Can you explain how the depolymerization occurs, which mechanism is involved? Or is it simple drifting of the sample in the CLSM due to thermal fluctuations next to the objectives of the microscope?

Comments on the data:

Question 7: Page 9, line 8: “A larger amount of peptide-type nanofibers were formed in the interstitial water space relative to those proximal to the lipid-type nanofiber surface resulting in the 10 predominant formations of the interpenetrated SDN.”

How do different ratios of the peptide and the lipid hydrogelator influence the gel formation?

Question 8: Page 9, line 19: Since CaOx-F(F)F is more hydrophilic at its N-terminal, it should exhibit poor hydrogelation compared with Ald-F(F)F according to our previously established design principle”.

As you assume that it should exhibit poor hydrogelation faculty, can you show a control experiment?

Question 9: Page 10, line 18, and supplementary Fig. 17 and figure 5b bottom right: “The line plot analysis demonstrated that the peak tops of peptide- and lipid-type nanofibers were not completely overlapped but slightly out of alignment (80 ± 30 nm), while the overall peak patterns were quite similar to each other (Fig. 5b, bottom right, supplementary Fig. 17).”

You explain that the overall peak patterns were quite similar and in figure 5b, bottom right, the patterns are quite similar. However, in supplementary figure 17, the patterns do not seem similar. The line plot for the interpenetrated SDN in figure 5b, top right, show alternated peaks, but the line

plot for the interpenetrated SDN in figure 3c (same compounds) shows no alternated peaks. This line plot looks more like the line plots in the supplementary figure 17 (of the parallel SDN).

Can you show a line plot for the interpenetrated SDN like for the parallel SDN in supplementary figure 17?

Also, the line plots are quite similar in all cases, as the lipid-type network is less dense compared to the peptide-type network, the probability to find fibers of the peptide network parallel to the lipid-network will happen. You might do the same observation in the interpenetrated network.

Does that mean that the kinetics of formation of the interpenetrated network is really fast and then as the network is really dense you didn't notice it?

Question 10: For the proof of the formation of the parallel network, you base it on the CLSM pictures, but to me, it's not sufficient proof as you might observe fibers parallel to the lipid-type network in the interpenetrated network. Also, the rheological properties cannot confirm the parallel aspect of the network, even if you observe that the hydrogel softens. As you are changing the composition of the fibers and the process seems to take time to "stabilize" at its final shape, you also disturb a pre-established network, to do so, it's not surprising that the hydrogel softens. As for most of the physical hydrogels, the first network is always the most robust one, when the gel is broken and reform, the mechanical properties are softer.

Question 11: Page 11, line 15: "These experiments indicated that the two-step oxime exchange was essential for the construction of the parallel SDN."

Can the same result be achieved without the first step (addition of CMA) by directly using CaOx-F(F)F and BHA?

Question 12: Your scheme figure 5 a is wrong according to previous figures and explanations. The interpenetrated network is obtained when you mix Ald-F(F)F and Phos-MecycC5 and you add the O-benzylhydroxylamine leading to the fast oxime formation giving rise to the interpenetrated network (figure 3a). So, how can you explain that you obtain a hydrogel in supplementary figure 22 when mixing Ald-F(F)F and Phos-MecycC5, then a solution when you add CMA and then the parallel network by addition of BHA.

The rheology measurement in figure 12a and c highlight it too that the solution containing Ald-F(F)F and Phos-MecycC5 is barely a gel according to your G' and G'' values, (Did you do the strain sweep first? Because the starting value of G' in the frequency sweep is almost 10 times lower compared to the strain sweep), and no fibers are observed for this solution in CLSM regarding the results presented in supplementary figure 6a. and 7.

Comments on the references:

The work of Adams and Tovar on the control over the self-sorting and coassembly of multichromophoric peptide hydrogelators (JACS 2017), and the work of Escuder and van Esch on the

tandem reactions in self-sorted catalytic molecular hydrogels (Chem. Sci. 2016) should be given, and the work of van Esch on the interpenetrating networks of fibers and surfactants (<https://doi.org/10.1039/B903806J>).

Reviewer #2 (Remarks to the Author):

In their manuscript, Hamachi and coworkers describe the controlled synthesis of multi-component hydrogels via in-situ generation of a peptide gelator. Careful tuning of the formation rate of the gelator allowed access to different morphologies of system. The conducted experiments were carefully performed and the in-situ imaging-based approach is beautifully used for this study. The results obtained are of great novelty and broad interest for the supramolecular community. I therefore recommend publication in Nature Communications after addressing the following points:

- 1) Many special words and abbreviations are used throughout the manuscript, which makes it not easy to read. For instance, it was difficult to discriminate between fiber formation and gel formation in the figures and text. It looked that they were used in a mixed way. Please have a look to make sure what is discussed at which point in the manuscript.
- 2) HPLC analysis was performed to monitor the reaction progress. Simple integration of the UV traces against an internal standard was used to estimate the concentrations of the peptide-aldehyde and the peptide-oxime. Yet, different extinction coefficients of the molecules are expected, which requires the need for a calibration curve to determine the concentration of the reaction components.
- 3) The experiments performed by the authors contain multiple components (two different types of oximes starting materials, peptide-aldehyde, two type of peptide-oximes, two types of fluorophores, Phos-MecycC5 gelator). As such small molecules can typical interfere with supramolecular assemblies, the effect of these components on the two gels formed should be evaluated. A systematic analysis of each component at different concentrations on the gel morphology is needed.
- 4) The rate of oxime formation/exchange can be tuned by using nucleophilic catalysts. How is the morphology of the gel affected, when the oxime exchange from CaOx-F(F)F to BnOx-F(F)F is accelerated? Is the selectivity reduced at higher reaction rates?
- 5) The two morphologies formed by fast and slow exchange is intriguing and explained by nucleation. Obviously, gels are kinetic traps, but it is also known that they change in time. So which one is now thermodynamically most favored (interpenetrating of parallel). Anyhow, some information about how the hydrogels behave in time would be very useful.

Minor comments:

- 6) Page 7, line 23, "To prove the importance.....": The experiments performed do not "prove" but rather "support" or "highlight" the importance of the authors findings.
- 7) Page 9, line 19: The word "terminal" should be replaced by "terminus".

Reviewer #3 (Remarks to the Author):

The manuscript by Kubota et al. reports the formation of supramolecular nanofibers consisting of peptide- and lipid-based gelator molecules, which results in self-sorting double network (SDN) hydrogels. A peptide-based molecule (Ald-F(F)F) can be converted to a better gelator molecule (BnOx-F(F)F) through dynamic covalent oxime chemistry, because of which the self-assembly process overall becomes under kinetic control. SDN hydrogel was obtained by performing such oxime formation for Ald-F(F)F in the presence of supramolecular nanofibers of the lipid-based gelator (Phos-MecycC5). Interestingly, two-types of self-sorted network patterns, interpenetrated and parallel network patterns, were obtainable by modulating the kinetics of dynamic covalent oxime chemistry. These distinct patterns were convincingly visualized by real-time CLSM imaging. The finding of the parallel self-sorting network is unprecedented, and kinetic control of the formation of distinct network patterns could lead to new supramolecular soft materials. The experiments are designed well. I recommend the publication of this manuscript in Nature Communications after the following points are addressed by the authors.

- 1) In page 9, line 1: The authors found two distinct processes in the formation of peptide-type nanofiber: one at the surface of the lipid-type nanofibers and the other in the interstitial water space, which occurred with slightly different kinetics. In the beginning of the oxime-formation, the former process was faster than the latter, however in the end, the latter type (i.e., penetrated networks) prevailed the system (Figure 4). Can the author explain why?
- 2) Related to the comment above, why did the authors expect that CMA could decelerate the nucleation in the interstitial water space selectively over the nucleation at the surface of the lipid-type nanofibers? Both the nucleation processes should be affected by the concentration of CMA.
- 3) In page 10: The interpenetrated SDN structure was obtained using Ald-F(F)F (17.3 mM) and Phos-MecycC5 (2.4 mM). It was a bit confusing in terms of “the importance of in situ peptide fiber formation (page 7, the last line)” that the interpenetrated SDN was obtained by heating-cooling process in this case (Methods). What is the difference between Ald-F(F)F and BnOx-F(F)F in the context of obtaining SDN network with Phos-MecycC5? Some kinetic effects?
- 4) The authors explained the difference between the oxime-formation and oxime-exchange protocols in terms of the rapid supersaturation state achieved in the former protocol. The supersaturated state then undergoes the nucleation-elongation process in a timeframe of 4 minutes (page 9 line 5). Supersaturation is inherently a kinetic effect, and there is other approach to avoid this. For example, if BHA was added gradually to a mixture of Ald-F(F)F and Phos-MecycC5; or if a hot solution of BnOx-F(F)F and Phos-MecycC5 was cooled slowly, it would be possible to obtain the

parallel self-sorting. I hope the authors, if possible, to comment on the general prerequisite or strategy to obtain the parallel self-sorting.

Reviewer #1 (Remarks to the Author):

In this manuscript, Ryou Kubota et al., describe a self-sorting double network hydrogels formation comprising orthogonal nanofibers peptide-type hydrogelator and lipid-type nanofibers. The authors describe that they can obtain two kind of self-sorting network, an interpenetrated network upon the addition of *O*-benzylhydroxylamine to a benzaldehyde-tethered peptide in the presence of the lipid-type nanofibers; and a parallel network upon a slow oxime exchange reaction from an oxime non-hydrogelator peptide-type to an oxime peptide-type hydrogelator in presence of the lipid-type hydrogelator. In the second case, the lipid- type hydrogelator play the role of seeding to obtain a parallel network formation.

The work claim novelty upon the first direct visualization of self-sorted network structures by using CLSM and the first control of pattern in situ of the network by slowing down the seeding kinetic through the use of oxime exchange chemistry to go from interpenetrated network to a parallel nanofibers network but it is rather preliminary for publication and the authors are hand-wavy the data assessment regarding the parallel network formation.

Conceptually, it may be valuable for Nat. Commun., however, I suggest to address the following comments:

Reply:

We appreciate your careful review and comments. We answered your concerns as shown below.

Question 1:

Page 3, line 14: “drug release matrices, drug delivery”. The two of them mean exactly the same thing.

Reply:

According to the reviewer’s comment, we modified the sentence as shown below.

Modification in the main text

Page 3, line 12:

The elaborate design of hydrogelators allows for construction of stimulus-responsive hydrogels, which are promising scaffolds for drug release matrices, ~~drug delivery~~, and

regenerative medicine.

Question 2:

The Ald-F(F)F form a gel on its own and the BnOx-F(F)F too, do the formation *in situ* of the BnOx-F(F)F change anything in the mechanical properties of the hydrogel? Also, do the BnOx-F(F)F hydrogel have similar hardness compare to the Ald-F(F)F hydrogel? How then the addition of a second self-sorted fibrous network changes the mechanical properties? Does the hydrogel soften or get harder?

Reply:

Supramolecular hydrogelators self-assemble into fibrous aggregates above the critical gelation concentration (CGC), whereas they remain the monomer state below CGC. Also, most of supramolecular hydrogelators tend to form precipitates at much higher concentration than CGC. Indeed, **BnOx-F(F)F** forms a precipitate after a heat-cool protocol at a higher concentration where **Ald-F(F)F** form a hydrogel (17.3 mM). We thus newly prepared the **BnOx-F(F)F** hydrogel *in situ* using **Ald-F(F)F** to answer the reviewer's comments and conducted rheological analysis. The storage modulus of the **BnOx-F(F)F** gel (8554 Pa) was much higher than that of the **Ald-F(F)F** gel (1629 Pa) (supplementary Fig. 4a–e). HPLC analysis showed that the oxime formation reaction proceeded in 68% (supplementary Fig. 4g). These results suggested that *in situ* formation of **BnOx-F(F)F** increased the mechanical property of the hydrogel, relative to **Ald-F(F)F** gel. We added these comments and data into the main text and supplementary information.

As shown in supplementary Fig. 3 and 16, the storage modulus of the self-sorted **BnOx-F(F)F/Phos-MecycC₅** network gel was 869 Pa, which was higher than the simple sum of **Phos-MecycC₅** sol and **BnOx-F(F)F** gel (32.5 and 418 Pa, respectively). Therefore, it is revealed that the integration of the peptide- and lipid-type network increased the mechanical property of the hydrogel. We added this discussion in the main text.

Modifications in the main text

Page 6, line 2:

HPLC analysis revealed that 95% of **Ald-F(F)F** was converted to **BnOx-F(F)F** (Fig. 2d). Also, the storage modulus of an **Ald-F(F)F** hydrogel (17.3 mM, above CGC) increased from 1629 to 8554 Pa upon treatment of *O*-benzylhydroxylamine

(supplementary Fig. 4).

Page 8, line 20:

Notably, the storage modulus of the interpenetrated SDN hydrogel (869 Pa) was higher than the simple sum of the **Phos-MecycC₅** solution and **BnOx-F(F)F** gel (32.5 and 418 Pa, respectively), suggesting that integration of the peptide- and lipid-type networks increased the mechanical property of the hydrogel.

Question 3:

Can you also control the pattern in the other way around, when the network of fibers is parallel can you transform it back to an interpenetrated one? Does it influence the hydrogel properties (mechanical or others)?

Reply:

We appreciate your important suggestion. We newly did the corresponding experiments and confirmed that **BnOx-F(F)F** nanofibers are stable and the oxime exchange reaction does not take place upon addition of an excess amount of carboxymethylamine to **BnOx-F(F)F**. The HPLC analysis demonstrated that 95% of **BnOx-F(F)F** (prepared by addition of 1 eq of *O*-benzylhydroxylamine to **Ald-F(F)F**) remained 24 h after treatment of an excess amount (10 eq) of carboxymethylamine (supplementary Fig. 38). Also, CLSM imaging showed that **BnOx-F(F)F** network did not seem to decompose by addition of carboxymethylamine. These data implied that the self-assembled fibrous structure of **BnOx-F(F)F** stabilizes the oxime bond of **Bn** against nucleophilic addition of carboxymethylamine probably due to the tight packing and/or the hydrophobic microenvironment. We added comments and experimental data into the main text and supplementary information as shown below.

Modification in the main text

Page 13, line 7:

Moreover, the direct treatment of the interpenetrated SDN of **Ald-F(F)F** and **Phos-MecycC₅** with *O*-benzylhydroxylamine never induced transformation to the parallel SDN (supplementary Figs. 35, 36). It was also confirmed that the further conversion from **BnOx-F(F)F** to **CaOx-F(F)F** did not proceed upon addition of an excess amount of carboxymethylamine, implying that the self-assembled structure stabilizes **BnOx-F(F)F** probably due to its tight packing and/or the hydrophobic

microenvironment (supplementary Fig. 38). These experiments indicated that the two-step oxime exchange was essential for construction of the parallel SDN.

Question 4:

Page 8, line 4: “In this case, a suspension containing a white precipitate was formed instead of a hydrogel (supplementary Fig. 11a), which was in sharp contrast with the *in situ* oxime-formation protocol that produced a slightly opaque hydrogel by addition of BHA to a viscous mixture of Ald-F(F)F and Phos-MecycC5 (supplementary Fig. 11b).”

- (1) What happens if you prepare the hydrogel by heating the two hydrogelators separately and then mix them together as in page 5 line 15 you seem to succeed to obtain a hydrogel for Ald-F(F)F and BnOx-F(F)F by heating the solution and let it cool down.
- (2) Can you explain why it works when you prepare an interpenetrated hydrogel from Ald-F(F)F and Phos-MecycC5 (like for the oxime exchange protocol with the conditions [Ald-F(F)F] = 17.3 mM (0.80 wt%), [Phos-MecycC5] = 9 2.4 mM (0.15 wt%)) and not when you mix BnOx-F(F)F and Phos-MecycC5 ?

Reply:

(1) We appreciate the reviewer’s comment. As suggested by the reviewer, we prepared a sample by heating two hydrogelator solutions separately and then mixing them together. In this case, however, we obtained a suspension containing white precipitates (supplementary Fig. 18a). CLSM imaging showed the heterogeneous mixture of **BnOx-F(F)F** and **Phos-MecycC₅** nanofibers (supplementary Fig. 18b). We thus concluded that *in situ* formation of **BnOx-F(F)F** through the oxime formation is promising to construction the homogenous self-sorting double network hydrogel. We modified the main text to explain these experimental results.

(2) According to our previous results [*Bioconjugate Chem.* **29**, 2058 (2018)], it is clear that one of the controlling factors over self-sorting phenomena is the optimal hydrophobicity of the peptide-type hydrogelators. We described therein that the peptide-type and lipid-type hydrogelators tend to form coassembled structures by the heat-cool protocol (for example, spherical aggregates), if a peptide-type hydrogelator is highly hydrophobic. The hydrophobicity of **BnOx-F(F)F** is higher than **Ald-F(F)F**. Therefore, **BnOx-F(F)F** gelator formed coassembled spherical aggregates with **Phos-MecycC₅** by the heat-cool protocol, while **Ald-F(F)F** and **Phos-MecycC₅** formed self-sorting nanofibers. To explain the self-sorting rules, we added the corresponding

sentences to the main text as shown below.

Modifications in the main text

Page 9, line 4:

A suspension containing white precipitates was also obtained by mixing hot solutions of **BnOx-F(F)F** and **Phos-MecycC₅** that were prepared separately (supplementary Fig. 18). Such coassembly behavior can be explained by the hydrophobicity of the peptide-type hydrogelator, one of the control factors over self-sorting phenomena we previously found.³⁷ If the peptide-type hydrogelator is highly hydrophobic, peptide- and lipid-type hydrogelators tend to form coassembled structures, such as spherical aggregates, by the heat-cool protocol.

Question 5:

Page 8, line 18: “dynamic covalent oxime chemistry”. What do you mean by dynamic? As your reaction is not auto-reversible how can you explain it? If it’s for the CLSM pictures showing the growth of fibers (Fig. 6, 7 and S5) can it be only dependent of the slow oxime exchange and slow growth of the fiber network, which may need several hours to reach it’s steady-state as the two chemical processes are kinetically different?

Reply:

Thank you very much for your comment. In page 8, line 18 (page 9, line 11 in the revised manuscript), we would like to claim here that “*in situ* formation of **BnOx-F(F)F** is important for formation of the self-sorting network.” We did not intend to discuss about (auto)reversibility of the oxime bond. As suggested by the reviewer, the slow oxime exchange and slow fiber elongation would be main factors to determine the kinetics in formation of the parallel SDN. We thus modified the sentence as shown below.

Modification in the main text

Page 9, line 11:

These results clearly indicated that kinetically controlled ~~*in situ*~~ formation of the hydrophobic **BnOx-F(F)F** using **the *in situ* oxime formation** is crucial for construction of the interpenetrated SDN.

Question 6:

Page 12, line 14: “Surprisingly, a few peptide-type nanofibers gradually collapsed (depolymerization) during the time-lapse imaging (Fig. 7e). The imaging data provided direct evidence that the formation of the artificial SDN structure is very dynamic and subject to thermal fluctuations”.

Does that mean that after some time the whole fiber network collapses? Can you explain how the depolymerization occurs, which mechanism is involved? Or is it simple drifting of the sample in the CLSM due to thermal fluctuations next to the objectives of the microscope?

Reply:

As shown in Fig. 5 and 6, the whole nanofiber network did not collapse and the sample kept the gel state until 48 h. In general, supramolecular fiber formation and degradation are reversible, because of the non-covalent self-assembly processes. The spatial heterogeneity of the hydrogel is provided in the transient state of the nanofiber formation, which may induce the locally-different temperature/viscosity and the locally-fluctuated monomer concentration. These would be contributed to the event that the nanofiber formation and collapse simultaneously proceed in the different spaces. Although the conventional spectroscopic measurements such as UV-vis, CD or fluorescence (providing the ensemble data) can't catch such events, *in-situ* observation using various microscopies may potentially address it. We believe our result is one of such cases. In Fig. 7e, we explained such stochastic behavior during the nanofiber network formation. Indeed, Sugiyasu *et al.* also visualized the similar dynamic behavior of nanofiber elongation and collapse during the seeded supramolecular polymerization by high-speed AFM (reference 71: *Angew. Chem. Int. Ed.* **57**, 15465 (2018)).

In the time-lapse imaging, we used the focus stabilization system (definite focus 2, Carl Zeiss) to compensate for the sample drift. Indeed, the surrounding peptide- and lipid-type nanofibers did not move during depolymerization (Fig. 7e). We added the explanation about the definite focus system into the supplementary method.

Modification in the supplementary information

Page 2, line 17:

For all of the time-lapse CLSM imaging, Definite Focus 2 (Carl Zeiss) was employed to compensate for the sample drift.

Question 7:

Page 9, line 8: “A larger amount of peptide-type nanofibers were formed in the interstitial water space relative to those proximal to the lipid-type nanofiber surface resulting in the 10 predominant formations of the interpenetrated SDN.”

How do different ratios of the peptide and the lipid hydrogelator influence the gel formation?

Reply:

According to the reviewer’s comment, we conducted the time-lapse CLSM imaging under different ratios of the peptide- and lipid-type hydrogelators. As shown in supplementary Fig. 13, the interpenetrated SDN structure was formed at a lower amount of **Ald-F(F)F** (2.2 mM) and a higher amount of **Phos-MecycC₅** (4.8 mM). Time-lapse movies also showed the nanofiber formation mechanism similar to the original concentration (supplementary Fig. 14). However, when using a lower concentration of **Phos-MecycC₅** (1.2 mM), we found spherical aggregates of **Phos-MecycC₅** mainly formed (supplementary Fig. 13b). We modified the main text and supplementary information as shown below.

Modification in the main text

Page 8, line 2:

It is also confirmed that a range of concentrations of the peptide-, lipid-type hydrogelators, and fluorescent probes scarcely affect formation of the interpenetrated SDN, except for a lower concentration of **Phos-MecycC₅** (supplementary Fig. 13 and 14).

Question 8:

Page 9, line 19: Since CaOx-F(F)F is more hydrophilic at its N-terminal, it should exhibit poor hydrogelation compared with Ald-F(F)F according to our previously established design principle”.

As you assume that it should exhibit poor hydrogelation faculty, can you show a control experiment?

Reply:

According to the reviewer’s comment, we determined the critical gelation

concentration (CGC) of **CaOx-F(F)F** to be 26 mM, which is much higher than that of **Ald-F(F)F** (8.6 mM) and **BnOx-F(F)F** (1.3 mM) (supplementary Fig. 19). These data indicated the hydrophobicity at the N-terminus is crucial for the hydrogelation property. We added a table of CGCs into SI (supplementary table 2), and modified the main text as shown below.

Modification in the main text

Page 10, line 14:

Since **CaOx-F(F)F** is more hydrophilic at its N-terminus, it should exhibit poor hydrogelation compared with **Ald-F(F)F** according to our previously established design principle.⁶⁷ Indeed, the critical gelation concentration of **CaOx-F(F)F** was determined to be 26 mM, which is much higher than that of **Ald-F(F)F** (8.6 mM) (supplementary Fig. 19, supplementary Table 2).

Question 9:

Page 10, line 18, and supplementary Fig. 17 and figure 5b bottom right: “The line plot analysis demonstrated that the peak tops of peptide- and lipid-type nanofibers were not completely overlapped but slightly out of alignment (80 ± 30 nm), while the overall peak patterns were quite similar to each other (Fig. 5b, bottom right, supplementary Fig. 17).”

You explain that the overall peak patterns were quite similar and in figure 5b, bottom right, the patterns are quite similar. However, in supplementary figure 17, the patterns do not seem similar. The line plot for the interpenetrated SDN in figure 5b, top right, show alternated peaks, but the line plot for the interpenetrated SDN in figure 3c (same compounds) shows no alternated peaks. This line plot looks more like the line plots in the supplementary figure 17 (of the parallel SDN).

Can you show a line plot for the interpenetrated SDN like for the parallel SDN in supplementary figure 17?

Also, the line plots are quite similar in all cases, as the lipid-type network is less dense compared to the peptide-type network, the probability to find fibers of the peptide network parallel to the lipid-network will happen. You might do the same observation in the interpenetrated network.

Does that mean that the kinetics of formation of the interpenetrated network is really fast and then as the network is really dense you didn't notice it?

Reply:

To show more clearly that the patterns in supplementary Fig. 17 (supplementary Fig. 24 in the revised manuscript) are similar to that in Fig. 5b bottom, we analyzed the peak patterns in more detail. We normalized the peak intensity at $5\ \mu\text{m}$ intervals so that the maximum and minimum intensities were set to 1 and 0, respectively. The analyzed patterns showed peak-tops of the peptide- and lipid-type nanofibers were almost identical but slightly misaligned. The average gap between peak-tops of the peptide- and lipid-type nanofibers were calculated to be $98 \pm 80\ \text{nm}$, which is almost identical to that of Fig. 5b bottom ($80 \pm 30\ \text{nm}$). Thus, it is concluded that the line plots in supplementary Fig. 17 (supplementary Fig. 24 in the revised manuscript) and Fig. 5b bottom show the similar pattern.

As suggested by the reviewer, we also analyzed the interpenetrated SDNs in Fig. 3b and Fig. 5b top like the parallel SDN (supplementary Fig. 9 and 21, respectively). These line plots showed that the peak patterns of the peptide- and lipid-type nanofibers seems different. To quantitatively evaluate the difference between the interpenetrated and parallel SDNs, we conducted statistical analysis of the peak-top distances. As shown in supplementary Fig. 25, the quantitative analysis demonstrated that the average peak-top distance of the parallel SDN ($98 \pm 80\ \text{nm}$) is statistically smaller than those of the interpenetrated SDNs ($180 \pm 150\ \text{nm}$ for Fig. 3b, $360 \pm 380\ \text{nm}$ for Fig. 5b top). The interpenetrated SDN formed by the oxime-formation protocol showed the intermediate value ($180 \pm 150\ \text{nm}$) because it was the mixture of the parallel and interpenetrated SDN as indicated by time-lapse imaging of formation process (Fig. 4).

To discuss the difference between the parallel and interpenetrated SDNs, we modified the main text as shown below.

Modification in the main text

Page 11, line 20:

The average peak-top distance of the parallel SDN was statistically smaller than those of the interpenetrated SDNs of **BnOx-F(F)F/Phos-MecycC₅** and of **Ald-F(F)F/Phos-MecycC₅** (see supplementary Fig. 25 for statistical analysis).

Question 10:

For the proof of the formation of the parallel network, you base it on the CLSM pictures, but to me, it's not sufficient proof as you might observe fibers parallel to the lipid-type network in the interpenetrated network. Also, the rheological properties

cannot confirm the parallel aspect of the network, even if you observe that the hydrogel softens. As you are changing the composition of the fibers and the process seems to take time to “stabilize” at its final shape, you also disturb a pre-established network, to do so, it’s not surprising that the hydrogel softens. As for most of the physical hydrogels, the first network is always the most robust one, when the gel is broken and reform, the mechanical properties are softer.

Reply:

No one has yet succeeded to give direct evidences for the parallel supramolecular network to date, and thus we here challenged to evaluate the parallel SDN by CLSM imaging. As answered in Question 9, our CLSM imaging can distinguish the two distinct fibers of the peptide- or lipid-type gelator and showed that these nanofibers formed the almost identical network pattern but slightly misaligned with each other. The peak-top distance was determined to be 98 ± 80 nm, which is within the spatial resolution of Airyscan (super-resolution) CLSM imaging. The Pearson’s coefficient that can quantify the degree of colocalization between two fluorescent probes is also regarded to be one of the appropriate indicators to judge the network patterns. From our results (including our previous papers), the Pearson’s coefficient value of the interpenetrated SDNs is typically below 0.3, indicating almost no correlation between the peptide- and lipid-type nanofibers (0.14 and 0.06 for Fig. 3b and Fig. 5b top, respectively). On the other hand, the parallel SDN gave the intermediate Pearson’s coefficient value to be around 0.4~0.5, suggesting that the peptide- and lipid-type nanofibers are moderately correlated (0.52 for Fig. 5b bottom). To assess the parallel SDN, the time-lapse CLSM imaging of the formation process is also carried out in this paper. Fig. 6 and supplementary movie 6 clearly demonstrated that 98.6% of peptide-type seeds formed on the surface of the lipid-type nanofibers, and the elongation direction from the seeds were identical to that of the lipid-type nanofibers (we confirmed that 277 out of 281 seeds were formed on and elongated along the lipid-type nanofibers). Moreover, 3D stacked images visualized that the peptide-type nanofibers were aligned to the lipid-type nanofibers, which is quite different from the interpenetrated SDNs (please compare supplementary movie 2, 4, 5). We believe that all of these data are sufficient for characterizing the parallel SDN network and conclude that our CLSM imaging is powerful to characterize the network patterns of SDN. We modified the main text as shown below.

We did not define the network patterns from the difference of rheological properties in this paper.

Modification in the main text

Page 13, line 20:

98.6% of peptide-type seeds (277 out of 281) were formed on the surface of the lipid-type nanofibers and elongated along the lipid-type nanofibers.

Question 11:

Page 11, line 15: “These experiments indicated that the two-step oxime exchange was essential for the construction of the parallel SDN.”

Can the same result be achieved without the first step (addition of CMA) by directly using CaOx-F(F)F and BHA?

Reply:

We appreciate the reviewer’s important suggestion. We newly conducted the experiments and confirmed that a suspension with white precipitates was obtained by the heat-cool protocol of **CaOx-F(F)F** and **Phos-MecycC₅** (supplementary Fig. 33). CLSM imaging visualized that the sample contained the heterogeneous **Phos-MecycC₅** nanofibers. These data implied that **CaOx-F(F)F** and **Phos-MecycC₅** interacted with each other probably because of the low hydrogelation property of **CaOx-F(F)F**. Thus, the initial formation process of **CaOx-F(F)F** is also essential for construction of the parallel self-sorting network. We added the results and comments into the main text and supplementary information.

Modification in the main text

Page 12, line 20:

The initial *in situ* formation of **CaOx-F(F)F** was essential for formation of the parallel SDN; a suspension with white precipitate was formed by the direct mixing of **CaOx-F(F)F** and **Phos-MecycC₅** with the heat-cool protocol (supplementary Fig. 33).

Question 12:

Your scheme figure 5 a is wrong according to previous figures and explanations. The interpenetrated network is obtained when you mix Ald-F(F)F and Phos-MecycC₅ and you add the O-benzylhydroxylamine leading to the fast oxime formation giving rise to the interpenetrated network (figure 3a). So, how can you explain that you obtain a

hydrogel in supplementary figure 22 when mixing Ald-F(F)F and Phos-MecycC5, then a solution when you add CMA and then the parallel network by addition of BHA.

The rheology measurement in figure 12a and c highlight it too that the solution containing Ald-F(F)F and Phos-MecycC5 is barely a gel according to your G' and G'' values, (Did you do the strain sweep first? Because the starting value of G' in the frequency sweep is almost 10 times lower compared to the strain sweep), and no fibers are observed for this solution in CLSM regarding the results presented in supplementary figure 6a. and 7.

Reply:

We are afraid that the reviewer may misunderstand the experimental conditions. Fig. 5a is correct. The concentration of **Ald-F(F)F** is different between Fig. 3a and 5a (4.3 and 17.3 mM, respectively: please also see the below table for critical gelation concentrations (CGC) of the hydrogelators). As replied in Question 2, the self-assembly behavior of supramolecular hydrogelators sharply changes at the critical gelation concentration. In the oxime-formation protocol (Fig. 3a), we adjusted the concentration of **Ald-F(F)F** (4.3 mM) lower than CGC (8.6 mM), and thus no peptide-type nanofibers were formed. After addition of *O*-benzylhydroxylamine, the self-sorting network was formed because the concentration of **BnOx-F(F)F** is above its CGC (1.32 mM). On the other hand, in Fig. 5 and supplementary Fig. 22 (supplementary Fig. 30 in the revised manuscript), because we set the **Ald-F(F)F** concentration (17.3 mM) higher than its CGC, it is reasonably expected that **Ald-F(F)F** self-assembled into nanofibers at the initial stage. When carboxymethoxylamine is added, it becomes a sol [the concentration is lower than the CGC of **CaOx-F(F)F** (26.1 mM)]. When *O*-benzylhydroxylamine is added to this solution, it becomes a gel because the concentration of **BnOx-F(F)F** is higher than its CGC.

In the case of supplementary Fig. 12a and 12c (supplementary Fig. 16 in the revised manuscript), the rheological behavior is reasonable because the concentration of **Ald-F(F)F** is below CGC (G' is higher than G'' due to the presence of the lipid fibers). We started the frequency sweep measurement followed by the strain sweep. The strain for the frequency sweep measurement was 1%. At this strain, the storage and loss moduli were nearly identical for the frequency and strain sweep measurements.

Table. Critical gelation concentrations of peptide-type hydrogelators.

	Ald-F(F)F	CaOx-F(F)F	BnOx-F(F)F
CGC (mM)	8.64	26.1	1.32

One-step oxime formation protocol: 4.3 mM

Two-step oxime exchange protocol: 17.3 mM

Comments on the references:

The work of Adams and Tovar on the control over the self-sorting and coassembly of multichromophoric peptide hydrogelators (JACS 2017), and the work of Escuder and van Esch on the tandem reactions in self-sorted catalytic molecular hydrogels (Chem. Sci. 2016) should be given, and the work of van Esch on the interpenetrating networks of fibers and surfactants (<https://doi.org/10.1039/B903806J>).

Reply:

Thank you very much for your suggestion. We newly cited three suggested papers.

Reviewer #2 (Remarks to the Author):

In their manuscript, Hamachi and coworkers describe the controlled synthesis of multi-component hydrogels via in-situ generation of a peptide gelator. Careful tuning of the formation rate of the gelator allowed access to different morphologies of system. The conducted experiments were carefully performed and the in-situ imaging-based approach is beautifully used for this study. The results obtained are of great novelty and broad interest for the supramolecular community. I therefore recommend publication in Nature Communications after addressing the following points:

Reply:

We really appreciate your careful reviewing and positive comments to our manuscript. To address your concerns, we amended the manuscript as shown below.

Comment 1:

Many special words and abbreviations are used throughout the manuscript, which makes it not easy to read. For instance, it was difficult to discriminate between fiber formation and gel formation in the figures and text. It looked that they were used in a mixed way. Please have a look to make sure what is discussed at which point in the manuscript.

Reply:

Thank you very much for your suggestion. According to your suggestion, we stop using abbreviations of *O*-benzylhydroxylamine (**BHA**) and carboxymethoxylamine (**CMA**). We added a table of abbreviations used in the manuscript into the supplementary information (supplementary Table 1). Besides, we modified the main text to discriminate between fiber and gel formation as shown below.

Modifications in the main text

Page 8, line 6:

To **support** the importance of *in situ* peptide fiber formation (termed oxime-formation protocol) by dynamic covalent chemistry for the interpenetrated SDN of **BnOx-F(F)F** and **Phos-MecycC₅**, we examined the self-sorting behavior of **BnOx-F(F)F** and **Phos-MecycC₅** **by macroscopic observation and rheological analysis.**

Page 10, line 21:

We conducted *in situ* imaging of fiber degradation and formation by the two-step oxime exchange process (see supplementary Fig. 20 for the staining selectivity).

Page 12, line 4:

We further analyzed the oxime-exchange process by HPLC analysis and the macroscopic phase transition.

Comment 2:

HPLC analysis was performed to monitor the reaction progress. Simple integration of the UV traces against an internal standard was used to estimate the concentrations of the peptide-aldehyde and the peptide-oxime. Yet, different extinction coefficients of the molecules are expected, which requires the need for a calibration curve to determine the concentration of the reaction components.

Reply:

In HPLC analysis, we calculated the conversion yields by comparing the integration between the same compounds. In supplementary Fig. 27, for example, the conversion rate of **CaOx-F(F)F** (72.2, 67.2, 70.3%) was estimated by use of the integration value of **CaOx-F(F)F** in the second row as 100%. To avoid readers' misunderstanding, we added the explanations to the figure captions.

Comment 3:

The experiments performed by the authors contain multiple components (two different types of oximes starting materials, peptide-aldehyde, two type of peptide-oximes, two types of fluorophores, Phos-MecycC5 gelator). As such small molecules can typical interfere with supramolecular assemblies, the effect of these components on the two gels formed should be evaluated. A systematic analysis of each component at different concentrations on the gel morphology is needed.

Reply:

According to the reviewer's comment, we newly investigated concentration dependence of peptide-, lipid-, fluorescent probes, and *O*-benzylhydroxylamine on formation of the interpenetrated and parallel SDNs. As shown in supplementary Fig. 13, in the one-step protocol, the interpenetrated SDN was successfully formed under a

range of concentrations of **Ald-F(F)F**, **Phos-MecycC₅**, **NBD-cycC₆**, **NP-Alexa647** (**Ald-F(F)F**: 2.2~4.3 mM, **Phos-MecycC₅**: 2.4~4.8 mM, **NP-Alexa647**: 4~10 μ M, **NBD-cycC₆**: 4~10 μ M). However, when we reduced the concentration of **Phos-MecycC₅** (1.2 mM), spherical aggregates mainly formed instead of nanofibers. In the two-step protocol, on the other hand, the parallel SDN was also formed under different concentrations we tested (**Ald-F(F)F**: 13~22 mM, **Phos-MecycC₅**: 2.4~4.8 mM, *O*-benzylhydroxylamine: 35~173 mM) (supplementary Fig. 32). We added these results to the main text and modified supplementary information to explain these additional experiments.

Modifications in the main text

Page 8, line 2:

It is also confirmed that a range of concentrations of the peptide-, lipid-type hydrogelators, and fluorescent probes scarcely affect formation of the interpenetrated SDN, except for a lower concentration of **Phos-MecycC₅** (supplementary Fig. 13 and 14).

Page 12, line 18:

The parallel SDN structure was successfully formed under different concentrations of **Ald-F(F)F**, **Phos-MecycC₅**, and *O*-benzylhydroxylamine (supplementary Fig. 32).

Comment 4:

The rate of oxime formation/exchange can be tuned by using nucleophilic catalysts. How is the morphology of the gel affected, when the oxime exchange from **CaOx-F(F)F** to **BnOx-F(F)F** is accelerated? Is the selectivity reduced at higher reaction rates?

Reply:

We appreciate the reviewer's intriguing advice. We newly conducted experiments to construct the parallel SDN in the presence of a nucleophilic aniline catalyst (response Fig. 1a). Unfortunately, however, a precipitate was formed when we added aniline to the single network composed of **CaOx-F(F)F** and **Phos-MecycC₅**. CLSM imaging after addition of *O*-benzylhydroxylamine with aniline (0.1 or 1.0 eq against the peptide-type hydrogelator) showed spherical aggregates stained by both peptide- and lipid-type probes (response Fig. 1b). We obtained the similar results when using other nucleophilic catalysts [methyl 3-amino-4-hydroxybenzoate or

2-(aminomethyl)benzimidazole] instead of aniline. These results indicated that these nucleophilic catalysts interacted with the peptide- and/or lipid-type nanofibers to induce coassembly of the hydrogelators.

To avoid interaction between hydrogelators and an aniline compound, we then sought to examine more hydrophilic aniline, sulfanilic acid (response Fig. 2a). In this case, however, the oxime exchange could not be accelerated by sulfanilic acid as confirmed by RP-HPLC (response Fig. 2b,c). CLSM imaging revealed that the parallel SDN structure was constructed in the presence of sulfanilic acid (Pearson's coefficient: 0.41) (response Fig. 2d,e). It implies that its nucleophilicity may not be sufficient for the oxime exchange.

As pointed out by the reviewer, any effects of nucleophilic catalysts on the network morphology may be indeed interesting, but we think that it is out of scope of this manuscript. We will carefully design nucleophilic catalysts and investigate their effects in the future.

Response Fig. 1. (a) Chemical structures of nucleophilic catalysts. (b) High-resolution Airyscan CLSM imaging (left) before addition, (second column from the left) after addition of carboxymethoxylamine, after addition of *O*-benzylhydroxylamine in the presence of (third column from the left) 0.1 or (right) 1.0 eq of an aniline catalyst. The aniline catalyst was added after treatment of carboxymethoxylamine. Condition: [Ald-F(F)F] = 17.3 mM (0.80 wt%), [Phos-MecycC₅] = 2.4 mM (0.15 wt%), [NP-Alexa647] = 4.0 μM, [NBD-cycC₆] = 4.0 μM, [carboxymethoxylamine] = 21 mM (1.2 eq), [O-benzylhydroxylamine] = 69 mM (4.0 eq), [aniline] = 1.73 or 17.3 mM in 100 mM MES, pH 6.0.

Response Fig. 2. (a) Chemical structure of sulfanilic acid. (b) HPLC analysis and (c) time course of the oxime exchange in the presence and absence of sulfanilic acid. (d) High-resolution Airyscan CLSM imaging of the parallel SDNs in the (top) presence and (bottom) absence of sulfanilic acid. (e) Line-plot analysis along a white line shown in response Fig. 2d. Condition: [Ald-F(F)F] = 17.3 mM (0.80 wt%), [Phos-MecycC₅] = 2.4 mM (0.15 wt%), [NP-Alexa647] = 4.0 μM , [NBD-cycC₆] = 4.0 μM , [carboxymethylamine] = 21 mM (1.2 eq), [O-benzylhydroxylamine] = 69 mM (4.0 eq), [sulfanilic acid] = 17.3 mM in 100 mM MES, pH 6.0.

Comment 5:

The two morphologies formed by fast and slow exchange is intriguing and explained by nucleation. Obviously, gels are kinetic traps, but it is also known that they change in time. So which one is now thermodynamically most favored (interpenetrating of parallel). Anyhow, some information about how the hydrogels behave in time would be very useful.

Reply:

Thank you very much for your valuable comment. According to the reviewer's suggestion, we monitored time-dependence of the interpenetrated and parallel SDNs by CLSM imaging. The interpenetrated SDN of **BnOx-F(F)F/Phos-MecycC₅** (prepared by the oxime-formation protocol) and **Ald-F(F)F/Phos-MecycC₅** did not show any significant changes at least for 3 days (supplementary Fig. 41a, 41b). In contrast, however, the parallel SDN gradually collapsed to form coassembled spherical aggregates after 3 days (supplementary Fig. 41c). Notably, we also found that the higher concentration of the peptide- or lipid-type hydrogelators suppressed collapse of the parallel SDN, suggesting that the stability of the peptide- and lipid-type nanofibers is critical for the kinetic trap of the parallel SDN (supplementary Fig. 41d, 41e). As the referee pointed out, these results implied that the parallel SDNs is the kinetically-trapped states and coassembled spherical aggregates is the thermodynamically-stable state. To explain these results, we modified the main text and supplementary information as shown below.

Modification in the main text

Page 15, line 23:

We monitored the stability of the interpenetrated and parallel SDNs by their time-dependent changes of CLSM images. The interpenetrated SDN of **BnOx-F(F)F/Phos-MecycC₅** (prepared by the oxime-formation protocol) and **Ald-F(F)F/Phos-MecycC₅** did not show any significant changes at least for 3 days (supplementary Fig. 41a, 41b). In contrast, the parallel SDN gradually collapsed to form coassembled spherical aggregates after 3 days (supplementary Fig. 41c). Interestingly, we found that the higher concentration of the peptide- or lipid-type hydrogelators suppressed collapse of the parallel SDN (supplementary Fig. 41d, 41e). These data suggested that the parallel SDN is the kinetically-trapped state and that the stability of the peptide- and lipid-type nanofibers is critical for the kinetic trap of the parallel SDN.

Comment 6:

Page 7, line 23, “To prove the importance.....”: The experiments performed do not “prove” but rather “support” or “highlight” the importance of the authors findings.

Reply:

According to the reviewer’s comment, we modified the main text as shown below.

Modification in the main text

Page 8, line 6:

To **support** the importance of *in situ* peptide fiber formation (termed oxime-formation protocol) by dynamic covalent chemistry for the interpenetrated SDN of **BnOx-F(F)F** and **Phos-MecycC₅**, we examined the self-sorting behavior of **BnOx-F(F)F** and **Phos-MecycC₅**.

Comment 7:

Page 9, line 19: The word “terminal” should be replaced by “terminus”.

Reply:

According to the reviewer’s comment, we modified the main text as shown below.

Modification in the main text

Page 10, line 14:

Since **CaOx-F(F)F** is more hydrophilic at its N-**terminus**, it should exhibit poor hydrogelation compared with **Ald-F(F)F** according to our previously established design principle.⁶⁷

Reviewer #3 (Remarks to the Author):

The manuscript by Kubota et al. reports the formation of supramolecular nanofibers consisting of peptide- and lipid-based gelator molecules, which results in self-sorting double network (SDN) hydrogels. A peptide-based molecule (Ald-F(F)F) can be converted to a better gelator molecule (BnOx-F(F)F) through dynamic covalent oxime chemistry, because of which the self-assembly process overall becomes under kinetic control. SDN hydrogel was obtained by performing such oxime formation for Ald-F(F)F in the presence of supramolecular nanofibers of the lipid-based gelator (Phos-MecycC5). Interestingly, two-types of self-sorted network patterns, interpenetrated and parallel network patterns, were obtainable by modulating the kinetics of dynamic covalent oxime chemistry. These distinct patterns were convincingly visualized by real-time CLSM imaging. The finding of the parallel self-sorting network is unprecedented, and kinetic control of the formation of distinct network patterns could lead to new supramolecular soft materials. The experiments are designed well. I recommend the publication of this manuscript in Nature Communications after the following points are addressed by the authors.

Reply:

We really appreciate your careful reviewing and positive comments to our manuscript. To address your concern, we amended the manuscript as shown below.

Comment 1:

In page 9, line 1: The authors found two distinct processes in the formation of peptide-type nanofiber: one at the surface of the lipid-type nanofibers and the other in the interstitial water space, which occurred with slightly different kinetics. In the beginning of the oxime-formation, the former process was faster than the latter, however in the end, the latter type (i.e., penetrated networks) prevailed the system (Figure 4). Can the author explain why?

Reply:

Thank you very much for your important question. We suppose that an amount of seeds formed in the interstitial water space is larger than those on the surface of the lipid nanofibers. As discussed in the nucleation of biological droplets formation, the nucleation barrier becomes lower when the system becomes supersaturated in a short

time, resulting in formation of many nuclei even at kinetically unfavored sites [reference 72: *Cell* **175**, 1481–1491 (2018)]. We considered that the situation is similar in the oxime-formation protocol, leading to formation of many nuclei at the interstitial water space. It is also possible that nanofiber elongation on the surface of the lipid-type nanofibers may be slower than at the interstitial water due to steric hindrance.

Comment 2:

Related to the comment above, why did the authors expect that CMA could decelerate the nucleation in the interstitial water space selectively over the nucleation at the surface of the lipid-type nanofibers? Both the nucleation processes should be affected by the concentration of CMA.

Reply:

We expected that the oxime exchange may slow down the generation kinetics of the **BnOx-F(F)F** molecule. Therefore, in the oxime-exchange process, the **BnOx-F(F)F** concentration would reach the supersaturated state much slower than the one-step protocol. It leads to preferential nucleation at the kinetically-favored site, the surface of the lipid-type nanofibers. To explain our hypothesis more clearly, we modified the main text as shown below.

Modifications in the main text

Page 10, line 8:

According to the *in situ* CLSM imaging data described above, we envisioned that deceleration of the ~~stochastic-seed **BnOx-F(F)F** generation in the interstitial water space~~ may enable preferential nucleation/elongation proximal to the lipid-type nanofibers leading to construction of a parallel self-sorting network.

Page 15, line 13:

In the oxime-formation protocol, the reaction system should reach a supersaturated state in a short time because of the fast generation kinetics of **BnOx-F(F)F**, therefore the nucleation proceeded in both sites. Additionally, the slower kinetics of ~~**BnOx-F(F)F** generation by the oxime exchange~~ allowed for preferential nucleation on the energetically-favorable surface of the **Phos-MecycC₅** nanofibers.

Comment 3:

In page 10: The interpenetrated SDN structure was obtained using Ald-F(F)F (17.3 mM) and Phos-MecycC5 (2.4 mM). It was a bit confusing in terms of “the importance of in situ peptide fiber formation (page 7, the last line)” that the interpenetrated SDN was obtained by heating-cooling process in this case (Methods). What is the difference between Ald-F(F)F and BnOx-F(F)F in the context of obtaining SDN network with Phos-MecycC5? Some kinetic effects?

Reply:

Thank you very much for your question. According to our previous results [*Bioconjugate Chem.* **29**, 2058 (2018)], one of the controlling factors over self-sorting phenomena is the optimal hydrophobicity of the peptide-type hydrogelators. We described therein that the peptide-type and lipid-type hydrogelators tend to form coassembled structures by the heat-cool protocol (for example, spherical aggregates), if a peptide-type hydrogelator is highly hydrophobic. The hydrophobicity of **BnOx-F(F)F** is higher than **Ald-F(F)F**. Therefore, **BnOx-F(F)F** gelator readily formed coassembled spherical aggregates with **Phos-MecycC₅** by the heat-cool protocol, while **Ald-F(F)F** and **Phos-MecycC₅** formed self-sorting nanofibers. To explain the self-sorting rules more clearly, we modified the main text as shown below.

Modifications in the main text

Page 9, line 6:

Such coassembly behavior can be explained by the hydrophobicity of the peptide-type hydrogelator, one of the control factors over self-sorting phenomena we previously found.³⁷ If the peptide-type hydrogelator is highly hydrophobic, peptide- and lipid-type hydrogelators tend to form coassembled structures, such as spherical aggregates, by the heat-cool protocol.

Comment 4:

The authors explained the difference between the oxime-formation and oxime-exchange protocols in terms of the rapid supersaturation state achieved in the former protocol. The supersaturated state then undergoes the nucleation-elongation process in a timeframe of 4 minutes (page 9 line 5). Supersaturation is inherently a kinetic effect, and there is other approach to avoid this. For example, if BHA was added gradually to a mixture of Ald-F(F)F and Phos-MecycC5; or if a hot solution of

BnOx-F(F)F and Phos-MecycC₅ was cooled slowly, it would be possible to obtain the parallel self-sorting. I hope the authors, if possible, to comment on the general prerequisite or strategy to obtain the parallel self-sorting.

Reply:

We really appreciate your interesting suggestion. According to your comment, we conducted stepwise addition of *O*-benzylhydroxylamine to a mixture of **Ald-F(F)F/Phos-MecycC₅** (0.2 eq each, 1.0 eq total). CLSM imaging revealed that both parallel and interpenetrated SDNs were formed, implying that control of nucleation sites is quite difficult by changing an amount of *O*-benzylhydroxylamine (supplementary Fig. 34). We also prepared a sample by slowly cooling down a hot solution of **BnOx-F(F)F** and **Phos-MecycC₅**. It resulted in the mixture of the interpenetrated and parallel SDNs (supplementary Fig. 35). We thus concluded that the oxime-exchange protocol is a promising way to selectively construct the parallel SDN. We added this result to the main text as shown below.

Modification in the main text

Page 12, line 23:

To decelerate the kinetics of seed formation in the oxime-formation protocol, we attempted stepwise addition of *O*-benzylhydroxylamine to a viscous solution of **Ald-F(F)F** and **Phos-MecycC₅**. In this case, however, the interpenetrated SDN was mainly formed (supplementary Fig. 34). Also, CLSM imaging of the sample prepared (a heterogeneous precipitate thus obtained) by slowly cooling down the hot solution of **BnOx-F(F)F** and **Phos-MecycC₅** showed the mixed network structure of interpenetrated and parallel SDNs (supplementary Fig. 35).

REVIEWERS' COMMENTS:

Reviewer #1 (Remarks to the Author):

I am satisfied with the adjustments the authors made and would like to congratulate them on a beautiful piece of work.

Reviewer #2 (Remarks to the Author):

I am very pleased with the careful revisions made by the authors and in the present form, I am even more excited to support the publication than I already was in the first round.

Reviewer #3 (Remarks to the Author):

The authors revised manuscript according to the reviewers' comments, and I now recommend the publication of this manuscript in Nature Communications.